 

# Task-anchored grid cell firing is selectively associated with successful path integration-dependent behaviour

Harry Clark, Matthew F Nolan*

Centre for Discovery Brain Sciences, Simons Initiative for the Developing Brain, Hugh Robson Building, University of Edinburgh, Edinburgh, United Kingdom

**Abstract** Grid firing fields have been proposed as a neural substrate for spatial localisation in general or for path integration in particular. To distinguish these possibilities, we investigate firing of grid and non-grid cells in the mouse medial entorhinal cortex during a location memory task. We find that grid firing can either be anchored to the task environment, or can encode distance travelled independently of the task reference frame. Anchoring varied between and within sessions, while spatial firing of non-grid cells was either coherent with the grid population, or was stably anchored to the task environment. We took advantage of the variability in task-anchoring to evaluate whether and when encoding of location by grid cells might contribute to behaviour. We find that when reward location is indicated by a visual cue, performance is similar regardless of whether grid cells are task-anchored or not, arguing against a role for grid representations when location cues are available. By contrast, in the absence of the visual cue, performance was enhanced when grid cells were anchored to the task environment. Our results suggest that anchoring of grid cells to task reference frames selectively enhances performance when path integration is required.

## eLife assessment

This **valuable** study examines the relationship between positional anchoring of grid cell activity and performance in spatial navigation tasks that require path integration. The authors demonstrate that grid cells can either fire in relation to the position relative to task-relevant virtual stimuli or independently based on the distance covered. Their findings **convincingly** reveal that mice exhibited better performance in the path integration task when grid cell activity was anchored to their position on the virtual track rather than the distance traversed, highlighting the contribution of grid firing to spatial navigation behavior. The work will be of interest to experimental and computational neuroscientists interested in spatial navigation.

**\*For correspondence:**
mattnolan@ed.ac.uk

**Competing interest:** The authors declare that no competing interests exist.

## Introduction

The ability to generate and manipulate internal representations of the sensory world is fundamental to cognitive functions of the brain. Grid representations generated by neurons in the medial entorhinal cortex (MEC) are thought to be critical for spatial behaviours and other cognitive functions that require structured representations (*Moser et al., 2008*; *Whittington et al., 2022*). However, the range of behaviours to which grid cells contribute is unclear (*Ginosar et al., 2023*). On the one hand, theoretical arguments that grid cell populations can generate high capacity codes imply that they could in principle contribute to all spatial behaviours (*Fiete et al., 2008*; *Mathis et al., 2012*;

*Sreenivasan and Fiete, 2011*). On the other hand, if the behavioural importance of grid cells follows from their hypothesised ability to generate position representations by integrating self-motion signals (*McNaughton et al., 2006*), then their behavioural roles may be restricted to tasks that involve path integration strategies.

Experiments that have manipulated circuits containing grid cells support the idea that they contribute to spatial behaviours. Lesions of the MEC disrupt place representations in the hippocampus and impair performance in spatial memory tasks (*Brun et al., 2008*; *Hales et al., 2021*; *Hales et al., 2018*; *Hales et al., 2014*; *Miao et al., 2015*; *Schlesiger et al., 2015*; *Steffenach et al., 2005*). These impairments are often incomplete in that hippocampal place representations remain, although they are less stable, and some spatial tasks can still be solved. More selective genetic manipulations support similar conclusions. Deletion during postnatal development of NMDA receptors from neurons in the MEC and nearby structures reduces the number of detected grid cells in these areas while having less effect on other spatial cell types (*Gil et al., 2018*). This manipulation impairs path integration without affecting other spatial behaviours (*Gil et al., 2018*). In contrast, targeted inactivation of stellate cells in layer 2 of the MEC, which are thought to be a major grid cell population (*Gu et al., 2018*; *Rowland et al., 2018*), impairs learning of both path integration-dependent behaviours and cue-based navigation more generally (*Qin et al., 2018*; *Tennant et al., 2018*). However, for all of these manipulations it is difficult to establish whether impairments result from deficits in grid firing per se, or from other alterations in the circuit and its potential for plasticity. It is also unclear whether functions that in these experiments appear resistant to perturbations of the MEC could have been restored by adaptive compensatory changes.

These challenges are common to efforts to test hypothesised functional roles for neural codes using perturbation strategies. A complementary approach is to take advantage of variability in the expression of behaviours and candidate neural representations. Thus, hypothesised functions for neural codes can be corroborated by correlations with behavioural outcomes, while dissociations between representations and behaviour may rule out hypothesised roles for a given code. In the case of grid codes, the idea that they provide a general-purpose spatial code predicts that they are always available and are stably anchored to the external environment (*Figure 1A*). This notion is challenged by observations that in circular track environments, grid firing patterns are maintained but are no longer anchored to the environment (*Jacob et al., 2019*). In this case, the grid representations are informative about distance travelled but not about absolute position (*Figure 1B*). Spatial representations within populations of MEC neurons that include grid cells can also be unstable with the network spontaneously remapping between different representations of location (*Low et al., 2021*). Thus, it appears that grid representations are not necessarily stably anchored to the external world, but it is unclear whether this instability impacts performance of spatial tasks. Here, we asked whether similar instability of grid activity manifests in a goal-directed task, and if so whether it can dissociate proposed behavioural roles for grid firing patterns.

We address this question by investigating grid and non-grid firing during a task in which mice learn the location of a reward on a virtual track (*Figure 1C*, *Tennant et al., 2022*; *Tennant et al., 2018*). On cued trials, the mice received rewards for stopping within a 'reward zone' that was marked by a distinct visual cue. In contrast to cue-rich virtual environments often used to study grid cells (e.g. *Campbell et al., 2018*; *Domnisoru et al., 2013*), this is the only spatially localised cue available after initiation of a trial. Removal of this cue enables testing of whether the mice are able to use a path integration strategy, in which case they should continue to selectively stop in the reward zone, or a cue-based strategy in which case they should no longer stop selectively at the reward zone location (*Figure 1C*; see also *Tennant et al., 2022*; *Tennant et al., 2018*). We find that during the task, grid cells can either be anchored to the track reference frame ('task-anchored'), or can maintain a periodic firing pattern independent of the track reference frame ('task-independent'). Adoption of these anchoring modes varied both between and within recording sessions. On trials when the reward zone cue was visible, adoption of the task-anchored representation did not predict task performance. By contrast, when the reward zone cue was absent, task-anchored grid firing was associated with successful localisation of the reward zone. Thus, our results suggest that task-anchoring of the grid cell network selectively enhances performance of behaviours that require path integration.



**Figure 1.** Models for grid representation and experiment design. (**A–B**) Predicted task-anchored (**A**) and task-independent (**B**) firing of grid cells in a 1D environment (right) given firing patterns of grid cells previously observed in square (**A**) and circular (**B**) 2D arenas (left). T1-T4 indicate consecutive trials in the 1D environment, AVG indicates expected average across many trials. (**C**) Neurons were recorded in an open arena and then in a location memory task. Trials were configured with a reward for stopping in a visually cued zone (beaconed), or a reward for stopping in the same zone but with the cue absent (non-beaconed), or without the visual cue or the reward (probe). Trial percentages indicate the proportion of trial types experienced in a single session; in any given session this proportion was fixed and trials were interleaved in a fixed repeating pattern (see Materials and methods). (**D**) In the task-anchored coding scheme, a grid cell fires with field spacing $\lambda$ and resets its firing every trial by anchoring its fields to the same track location, with a realignment lag R observed in the spatial autocorrelogram. Fields locations remain constant on each trial and thus peaks in the periodogram occur at integer spatial frequencies relative to the track repetition (see *Figure 1—figure supplement 3*). (**E**) In a task-independent coding scheme, a grid cell fires with field spacing $\lambda$ and continues to fire at regular intervals without anchoring to the track. Unless field spacing and the track length are integer divisible, the location of fields varies across trials, and thus the peak of the periodogram is not constrained to an integer spatial frequency (see *Figure 1—figure supplement 3*).

The online version of this article includes the following figure supplement(s) for figure 1:

**Figure supplement 1.** Procedure for extracting a session-level periodogram from a set of spike timestamps.

**Figure supplement 2.** Procedure for estimating a false alarm threshold for a given cell.

**Figure supplement 3.** Identification of task-anchored and task-independent periodic firing.

**Figure supplement 4.** Expected spatial periodicity for different functional cell types.

**Figure supplement 5.** Validation of accuracy and bias for classification of task-anchored and task-independent modes.

## Results

We recorded from neurons in the MEC of nine wild-type mice exploring an open arena and then performing a location memory task in a virtual linear track environment. By comparing the hexagonal symmetry of spatial autocorrelograms of neural activity in the open arena with corresponding shuffled data, we identified 103/1881 neurons as grid cells (11.4±17.4 grid cells/mouse, range 0.4–9.5%, *Supplementary file 1*) (see Materials and methods for classification procedures). These grid cells had field sizes of 7.5±3.3 cm and grid spacing of 72.5±13.7 cm and were found in dorsomedial parts of the MEC. Until indicated otherwise, we report analyses of neurons across all trials of the location memory task regardless of whether the reward zone is indicated by the cue or whether mice stop at the rewarded location.

### Grid cells exhibit either task-anchored or task-independent firing

A priori, we envisaged two scenarios for activity of grid cells during the location memory task. Given well-established spatial firing of grid cells in open arenas, we might expect that grid cell activity is anchored to the task reference frame (*Figure 1A and D*). Alternatively, given distance encoding but location-independent firing of grid cells in circular tracks (*Jacob et al., 2019*), we might expect the activity of grid cells to be periodic but independent from the task reference frame (*Figure 1B and E*).

To distinguish these scenarios we estimated the periodicity of each neuron's activity as a function of distance moved by the mouse using the Lomb-Scargle method (*Lomb, 1976*; *Scargle, 1982*; *VanderPlas, 2018*, *Figure 1—figure supplements 1–3*). This approach yields periodograms indicating power as a function of oscillation frequency and an associated estimate of the false alarm probability. We validated these estimates for detection of periodicity associated with grid firing using synthetic and shuffled data (*Figure 1—figure supplements 4 and 5*). According to the task-anchored firing scheme, firing fields should occur at the same positions on each trial (*Figure 1A*). In this case, because the virtual track repeated every 200 cm, significant peaks in the periodogram should occur at integer multiples of the spatial frequency of the repeating track (*Figure 1D*, *Figure 1—figure supplement 3*). By contrast, in the task-independent firing scheme, grid representations do not anchor to the track, but maintain firing fields that are periodic with respect to distance run (*Figure 1B*). In this case, significant peaks in the periodogram would reflect the distance between firing fields repeating independently of the task reference frame (*Figure 1E*, *Figure 1—figure supplement 3*). A further possibility is that in the location memory task the activity of grid cells is no longer periodic. In this case, peaks in the periodogram above the false alarm threshold should be absent.

We initially tested these predictions for neural activity across a complete behavioural session (n=103 grid cells, N=61 sessions, 233±135 trials/session)(*Figure 2*). We found that 68 of 103 grid cells had peaks in their periodograms within 5% of an integer multiple of the spatial frequency of the track repetition, consistent with their activity being anchored to the reference frame provided by the virtual track (*Figure 2A and D–E*). We refer to these neurons as showing 'session-level task-anchored' grid firing. By contrast, 32 of 103 grid cells had periodograms with peaks at frequencies outside 5% of an integer multiple of the track length, indicating that their activity was not coupled to the task reference frame (*Figure 2B and D–E*). We refer to these neurons as showing 'session-level task-independent' grid firing. Grid cells with periodograms lacking peaks above the false alarm threshold, which we will refer to as aperiodic grid cells, were rare (3/103) (*Figure 2C–E*). When we carried out similar analyses for non-grid cells, we found that the proportion of neurons with task-anchored firing was similar, but task-independent periodic firing was rarer, while aperiodic firing neurons were more common (*Figure 2E*).

To validate the periodogram-based classification we calculated the mean firing of grid cells as a function of track position. Consistent with their classification, task-anchored grid cells showed clear firing rate peaks associated with specific track locations (*Figure 2A*). By contrast, task-independent and aperiodic grid cells lacked clear firing rate peaks (*Figure 2B and C*). These differences manifest as substantially higher spatial information scores, but similar average firing rates, for grid cells with session-level task-anchored firing compared with task-independent firing (*Figure 2F*). Features of the periodogram such as peak power and peak width did not differ between task-anchored grid cells and task-independent grid cells (*Figure 2F*). The aperiodic grid cells had much lower mean firing rates than the task-anchored grid or task-independent grid cells suggesting that their apparent lack of periodicity reflects inactivity (*Figure 2F*).

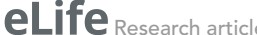

**Figure 2.** Grid cells operate in task-anchored or task-independent firing modes. (**A–C**) Examples of grid cells with activity during the location memory task classified as task-anchored (**A**), task-independent (**B**), or aperiodic (**C**) at the session level. Examples are ordered in their respective groups by their spatial information on the track. From top to bottom plots show: heap map of firing rate as a function of track position, spatial autocorrelation of the track firing rate, periodogram of the track firing, open field firing rate map, and open field spatial autocorrelogram. The red line indicates the false alarm threshold estimated from shuffled data and significant peaks are labelled with a triangle. X-axis scales are adjusted on the virtual reality spatial autocorrelation to better illustrate the long-range periodic signal. (**D**) Peak power as a function of the spatial frequency at which the peak occurs for all recorded cells. The red dashed line indicates the false alarm threshold generated from shuffled data. (**E**) Percentage of grid (G) and non-grid (NG) cells classified to task-anchored, task-independent, and aperiodic groups. (**F**) Comparison between task-anchored (TAG), task-independent (TIG), and aperiodic (AG) grid cells of mean firing rate (ANOVA: DF=2, p=0.006, $X^2$=10.215; pairwise comparisons: TAG vs TIG, DF=4.12, p=0.8, T=0.562; TAG vs AG, DF=42.07, p=0.02, T=2.906; TIG vs AG, DF=67.29, p=0.03, T=2.588), spatial information scores (ANOVA: DF=2, p=0.008, $X^2$=9.54; pairwise comparisons: TAG vs TIG, DF=73.6, p=0.02, T=2.815; TAG vs AG, DF=95.5, p=0.11, T=2.036; TIG vs AG, DF=96.6, p=0.7, T=0.783), peak power (ANOVA: DF=2, p=0.001, $X^2$=13.792; pairwise comparisons: TAG vs TIG, DF=19.0, p=0.13, T=2.033; TA vs A, p=0.006, T=3.239, DF=54.9; TI vs A, p=0.07, T=2.274, DF=80.4), and peak width (ANOVA: DF=2, p=0.15, $X^2$=3.7963; pairwise comparisons: TAG vs TIG, p=0.61, T=−1.029, DF=3.27; TA vs A, p=0.88, T=0.472, DF=45.76; TI vs A, p=0.55, T=1.053, DF=71.22).

These data suggest that during the location memory task we consider here, grid firing can either be anchored to the track and therefore be directly informative about position relative to the task reference frame, or be independent of the track and therefore may only be directly informative about distance travelled within the behavioural reference frame. Anchoring is consistent with previous reports of grid cell activity on virtual and real-world linear tracks (**Domnisoru et al., 2013**), while

task independence is consistent with encoding of distance but not position by grid cells in real-world circular tracks (*Jacob et al., 2019*).

## Grid cells switch between task-anchored and task-independent firing

We next asked if the mode adopted by the grid cells at the level of a whole session, either task-anchored or task-independent, was stable across individual trials within the session (*Figure 3A*, left), or if cells could switch mode (*Figure 3A*, right). Visual inspection of firing rate heat maps indicated that for some grid cells their firing pattern was stable across most trials within a session (*Figure 3B and C*). However, for many grid cells there appeared to be clear changes in anchoring within a session (*Figure 3D and E*). These switches could not be explained by variation between trials in the availability of cues or rewards, as these were interleaved in blocks that repeated throughout a session (see Materials and methods), whereas periods in which grid cell activity was in a given mode extended across the repeating blocks (e.g. *Figures 3D, E, 4A, 5E, F*).

To quantify switching between firing modes we evaluated rolling periodograms across each session. We classified each periodogram window as task-anchored, if the periodogram peaks occurred at integer multiples of the spatial frequency at which the track repeats, as task-independent if there were periodogram peaks at other spatial frequencies, or as aperiodic if there were no peaks above the false alarm threshold (see Materials and methods and *Figure 3—figure supplements 1–3*). For 26.2% of grid cells (27/103) the classification was consistently (>85% of trials) task-anchored or task-independent (*Figure 3F*), whereas for 73.8% of grid cells (76/103) no single coding scheme accounted for more than 85% of trials (*Figure 3F*). Grid cells identified as having task-anchored firing in our initial session-level analysis (*Figure 2*) were biassed towards high proportions of task-anchored trials (*Figure 3B and F*), whereas grid cells identified as task-independent at the session-level showed a bias towards high proportions of task-independent trials (*Figure 3C and F*), although in both groups many neurons showed variation between trials indicating that session-level analyses may obscure dynamic changes in task-anchoring. Differences in spatial information and periodogram properties between grid cells classified as task-anchored or task-independent based on their session-level firing patterns (*Figure 2F*) were nevertheless maintained when we instead compared neurons with firing mode that was consistent across trials within a session (*Figure 3—figure supplement 4*). For non-grid cells, consistent with session-level analyses, we again found that a sub-population showed task-anchored firing at a trial level, while task-independent periodic firing was substantially less common than in the grid cell population (*Figure 3—figure supplement 5*).

To validate the window-based analyses we compared mean firing rate as a function of track position between trials classified as task-anchored and those classified as task-independent. Consistent with the classification scheme correctly separating task-anchored from task-independent activity, spatially localised firing rate fields were present during task-anchored firing but were reduced or abolished when firing was task-independent (e.g. see rate maps, *Figure 3D and E*). This difference was associated with substantially lower spatial information scores for trials with task-independent firing, while mean firing rates were similar (*Figure 3G*). As a further test, we compared the durations of periods of activity in a given state to durations generated from shuffled datasets. If classification was by chance then the distributions of durations should be similar. By contrast, among all grid cells periods of task-anchored or task-independent firing extended in blocks across multiple trials, with the distribution of block lengths substantially different to that generated by shuffled data (*Figure 3—figure supplement 6*)(p=0.0003, KS=0.066; DF=1765, 2359; Kolmogorov-Smirnov test). Variation between grid cells in their firing mode also could not be explained by the relative proportion of cued trials in a session (*Figure 3—figure supplement 7*).

Given that populations of grid cells show coordinated dynamics that are consistent with their forming networks with continuous attractor dynamics (*Barry et al., 2007*; *Waaga et al., 2021*; *Yoon et al., 2013*), changes in anchoring should be coherent across grid cells and any non-grid cells that coordinate with the grid network. To test this, we compared activity patterns of simultaneously recorded grid and non-grid cells (*Figure 4*). Grid cells typically transitioned between task-anchored and task-independent firing modes at the same time (e.g. see *Figure 4A–C* and *Figure 5E and F*). This manifested as strong agreement between grid cells in their session-long order of task-anchored and task-independent epochs (*Figure 4D and E*). By contrast, the activity of non-grid cells was more diverse. Some non-grid cells transitioned between task-anchored and task-independent modes at the



**Figure 3.** Coding schemes switch within behavioural sessions. (**A**) In a 'stable coding' scenario grid cells remain either task-anchored or task-independent throughout the recording session (left), whereas with 'unstable coding' the grid activity switches between task-anchored and task-independent modes (right). (**B–E**) Example trial-by-trial firing rate heat maps (upper left), corresponding rolling periodogram heat maps (upper right), mean rate maps (lower left), and mean periodograms (lower right) for neurons exhibiting stable task-anchored coding (**B**), stable task-independent

*Figure 3 continued*

coding (**C**), and unstable coding in which representations switch between task-anchored and task-independent (**D–E**). (**F**) Distribution across all recorded grid cells of task-anchored trials (left), task-independent trials (centre), and aperiodic trials (right). Session-level task-anchored grid (TAG), task-independent grid (TIG), and aperiodic grid (AG) cell classifications are differently coloured. (**G**) Spatial information was higher for trials when grid cells were task-anchored compared to when they were task-independent (left) (ANOVA: p<1e-7, $X^2$=30.98, DF=1), whereas the average firing rate was similar between task-anchored and task-independent trials (right) (ANOVA: p=0.88, $X^2$=0.022, DF=1).

The online version of this article includes the following figure supplement(s) for figure 3:

**Figure supplement 1.** Procedure for classifying periodicity on a rolling basis.

**Figure supplement 2.** Evaluation of classification accuracy and bias as a function of the number of periodograms used for rolling classification.

**Figure supplement 3.** Classification accuracy and bias on the level of trial as a function of rolling window size.

**Figure supplement 4.** Spatial firing properties of stable task-anchored and task-independent grid cells.

**Figure supplement 5.** Grid cells exhibit task-independent periodic codes more frequently than non-grid cells.

**Figure supplement 6.** Assessment of the length of periodic coding blocks.

**Figure supplement 7.** Differences in order and ratio of trial types do not explain the variability in task-anchoring.

same time as simultaneously recorded grid cells, whereas others remained stably anchored throughout the recording session (*Figure 4A–C* and *Figure 5E and F*). As a result, session-long agreement scores for pairs of grid and non-grid cells were more variable and on average lower than for pairs of grid cells (*Figure 4D and E*). Non-grid cells also differed from the grid cell population in that their activity maintained similar spatial information between trials when grid cells were task-anchored versus when they were task-independent, whereas for grid cells task independence was associated with reduced spatial information (*Figure 4F*).

Together, these data indicate that grid cells can switch between task-anchored and task-independent firing modes within a behavioural session. This switching happens coherently across grid cell populations, which is consistent with grid cells forming networks with continuous attractor dynamics (*Barry et al., 2007*; *Gardner et al., 2022*; *Waaga et al., 2021*; *Yoon et al., 2013*). Our data also suggest that non-grid cells within the MEC form multiple populations, with some having activity that is coherent with the grid cell network, whereas others do not show task-independent periodic firing but instead maintain stable spatial representations independently from grid cells.

## Task-anchored coding by grid cells is selectively associated with successful path integration-dependent reward localisation

Our analyses indicate that grid cells exhibit either task-anchored or task-independent firing, and that their activity can switch between these modes within a recording session. Since task-anchored firing fields could be read out directly to estimate track location, but location may only be inferred indirectly from task-independent activity, we reasoned that the presence or absence of task-anchoring could be used to assess whether grid firing contributes to the ongoing behaviour. Because in grid networks the activity of an individual neuron is informative about the network state as a whole (*Fiete et al., 2008*; *Gardner et al., 2022*; *Waaga et al., 2021*), then in principle activity of any grid cell is indicative of whether the grid network as a whole is in a task-anchored or task-independent mode (see also *Figure 4*). Thus, if anchoring of grid representations to the task environment is critical for localisation of the reward, then task-anchored coding of individual neurons should predict successful trials. On the other hand, if behavioural performance is maintained when the grid representation is task-independent, then it is unlikely that anchored grid representations are necessary for reward localisation.

To distinguish these possibilities we took advantage of variation in behavioural outcomes, which were such that mice either stopped correctly in the reward zone ('hit' trials), slowed down on approach to the reward zone but did not stop ('try' trials), or maintained a high running speed across the reward zone ('run' trials)(*Figure 5A and B* and *Figure 5—figure supplements 1 and 2*). We separately evaluated outcomes from trials in which the reward zone cue was visible and a reward available ('beaconed trials'), trials in which the reward zone cue was omitted and a reward was available ('non-beaconed trials'), and trials in which the cue and the reward were both omitted ('probe trials'). Because trials of each type were interleaved into blocks that were repeated across a session (see Materials and methods), while periods of task-anchored and task-independent activity were typically maintained for



**Figure 4.** Grid cells and non-grid cells switch between coding schemes coherently. (**A**) Joint activity of 6 simultaneously recorded grid cells (orange frames) and 18 non-grid cells (blue frames) from a single session. For each cell, the firing rate map across trials (left) is shown next to the trial classification (right). (**B**) Classifications for all grid cells (GC) and non-grid cells (NG) as shown in (**A**), ordered by their agreement to the most common classification within the recorded network at any particular trial. The common classification for recorded grid cells is shown as $\overline{G}$ . (**C**) Mean firing rate as a function of position for exemplar units from (**A**) when $\overline{G}$ was task-anchored (left) and task-independent (right). (**D**) Strategy for assessing agreement between cells in their firing mode (i) and for generating shuffled datasets (ii). (**E**) Agreement in the firing mode between each combination of grid (G) and non-grid (NG) cells and corresponding scores for the shuffled data (lower), and the difference between the shuffled and actual scores (upper). Agreement was greater between grid cell pairs than between pairs involving non-grid cells (ANOVA: G-G, p<1e-13, $X^2$=58.89, DF =1, NG-NG, p<1e-5, $X^2$=20.42, DF =1, G-NG, p<1e-7, $X^2$=30.37, DF =1; pairwise comparisons: G-G vs NG-NG, p<1e-4, T=–10.455, DF =10,720, G-G vs G-NG, p<1e-4, T=–8.415, DF =10,710, NG-NG vs G-NG, p<1e-4, T=–6.853, DF =10,352). (**F**) Spatial information of individual cells during trials in which $\overline{G}$ is task-independent as a function of spatial information during trials in which $\overline{G}$ is task-anchored (left). The difference in spatial information between sessions classed as task-anchored or task-independent on the basis of grid cell activity was greater for grid than non-grid cells (right, ANOVA: p<1e-5, $X^2$=21.1, DF =1; G vs zero, p=0.018, T=2.723, DF =11.87; NG vs zero, p=0.3, T=–1.278, DF =2.58). The percentage change in spatial information was calculated as $100 \cdot \left( SI_{\overline{G}=TA} - SI_{\overline{G}=TI} \right) / SI_{\overline{G}=TA}$ .



**Figure 5.** Spatial behaviour during task-anchored and task-independent grid modes. (**A**) Averages across each behavioural session of running speeds as a function of track position for all sessions. (**B**) Running speed as a function of track position for trial outcomes classified as hit, try, or run for an example session. (**C–F**) Examples of variation in the behaviour-related activity of grid and non-grid cells recorded on the location memory task, illustrating firing patterns that are stable and task-anchored (**C**) or task-independent (**D**) firing, and unstable firing where cells switch between task-anchored and task-independent modes (**E–F**). Plots show all simultaneously recorded cells' firing rate maps in each session (left), stop rasters (lower centre), stop density on beaconed (B) and non-beaconed (NB) trials coloured according to whether grid cells were task-anchored or task-independent (upper centre) and

*Figure 5 continued on next page*

*Figure 5 continued*
a summary of raster of behaviour and cell classifications (right). Shaded regions in stop density plots represent standard error of the mean measured across epochs. The number of trials classified in a particular coding scheme is also provided with the stop density plot. Grid cells and non-grid cells are colour-coded by bounding boxes around the firing rate map.

The online version of this article includes the following figure supplement(s) for figure 5:

**Figure supplement 1.** Classification of trial outcomes.

**Figure supplement 2.** Speed profiles across behavioural sessions for each mouse.

**Figure supplement 3.** Further examples of spatial behaviour during task-anchored and task-independent grid modes.

blocks of many trials, differences in behavioural outcomes could not be explained by association of the grid-anchoring mode with particular trials types.

We first compared stopping behaviour when grid cells showed task-anchored firing that was stable within a session to when they showed task-independent firing that was stable within a session. On cued trials the spatial organisation of stopping behaviour (*Figures 5C vs D and 6A*) and the proportion of hit trials (*Figure 6B*) was similar for both groups. By contrast, on non-beaconed and probe trials stopping behaviour was clearly spatial when grid activity was task-anchored, but spatial organisation was largely absent when grid activity was task-independent (*Figures 5C vs D and 6A*), while the proportion of hit trials was substantially reduced (*Figure 6B*). These observations are consistent with grid representations being required for path integration-dependent but not cued localisation of the reward zone. This session-level analysis has the advantage that it focuses on large blocks of time (sessions) during which the mode of grid firing was stable, but the disadvantage that it excludes many sessions in which the mode of grid firing switches between task-anchored and task-independent.

We addressed this by using additional trial-level comparisons to evaluate all behavioural sessions, including those when the grid mode was unstable. On cued trials the spatial organisation of stopping behaviour (*Figures 5E–F and 6C*), and the proportion of hit trials (*Figure 6D*), was similar irrespective of whether grid cell fields were task-anchored or task-independent. On trials in which the reward zone cue was hidden, the relationship between firing and behavioural outcome was more complex. In many sessions, localisation of the reward occurred almost exclusively when cell firing was task-anchored and not when it was task-independent (*Figure 5E and F*). In a few sessions, we observed spatial stopping behaviour comparable to cued trials, even when grid firing was predominantly task-independent (*Figure 5—figure supplement 3*). In one session, we also observed grid cells shifting phase between task-anchored epochs (*Figure 5E*) similar to findings in *Low et al., 2021*, although this did not appear to alter task performance. On average, spatial stopping behaviour was reduced on non-beaconed and probe trials during which grid firing was task-independent (*Figure 6C*), and the proportion of successful trials was reduced compared with when grid firing was task-anchored (*Figure 6D*). The differences in outcomes between task-anchored versus task-independent trials were not associated with differences in running speed profiles (*Figure 6—figure supplement 1*) indicating they were not a consequence of a difference in motor behaviour.

Additional analyses of relationships between firing mode and trial outcomes were consistent with these observations. Thus, task-anchored firing occurred on a greater proportion of non-beaconed and probe trials that were hits compared with task-independent firing, but on a smaller proportion of run trials, in which mice ran through the reward zone without slowing down (*Figure 6E*). In addition, the odds ratio for receipt of reward on task-anchored versus task-independent trials was close to 1 when cues were available, but was much larger in the absence of cues (*Figure 6F*). Analyses using an alternative classification method, based on template matching of the task-anchored firing rate profile (*Figure 6—figure supplement 2A-B*), also indicated a similar behavioural performance on cued trials, but impaired performance on uncued trials when grid cells were in a task-independent mode (*Figure 6—figure supplement 2C-F*).

Together, these analyses demonstrate that anchoring of grid codes to track position is not necessary to successfully obtain rewards when visible cues are present. By contrast, when visual cues that indicate the reward zone are absent, task-anchored coding appears to promote successful localisation.

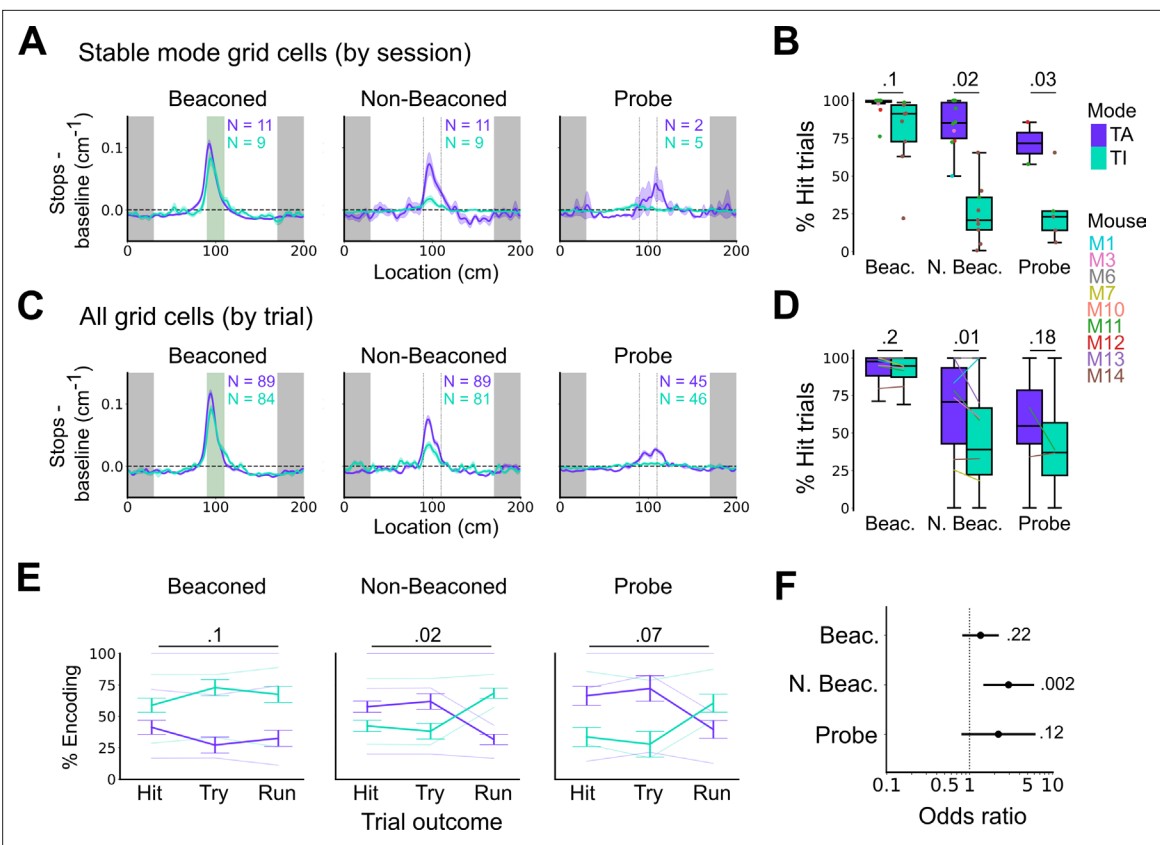

**Figure 6.** Anchoring of grid firing to the task reference frame promotes localisation by path integration but is not required for cued localisation. (**A, C**) Stopping probability relative to baseline as a function of position for sessions with stable grid codes (**A**) and for epochs within sessions containing both stable and unstable codes (**C**). Shaded regions in A and C represent standard error of the mean measured across sessions and epochs respectively. N represents the number of sessions with stable codes (**A**) and the number of grid cells with coding epochs (**C**). Stable sessions were defined as a session with at least one grid cell for which >85% of trials were in a single coding mode. (**B, D**) Percentage of hits on beaconed, non-beaconed, and probe trials when the code is task-anchored or task-independent for (**B**) sessions with stable codes (ANOVA: beaconed, p=0.1, $X^2$=2.70, DF=1; non-beaconed, p=0.022, $X^2$=5.24, DF=1; probe, p=0.033, $X^2$=4.55, DF=1) and when epochs within sessions are task-anchored or task-independent (**D**) (ANOVA: beaconed, p=0.09, $X^2$=2.84, DF=1; non-beaconed, p=0.001, $X^2$=10.14, DF=1; probe, p=0.19, $X^2$=1.70, DF=1, see Materials and methods). (**E**) Percentage of trials with hit, try, and run outcomes in which grid cell firing is task-anchored (TA) or task-independent (TI) (ANOVA: beaconed, p=0.11, $X^2$=4.49, DF=2; non-beaconed, p=0.02, $X^2$=7.75, DF=2; probe, p=0.07, $X^2$=5.40, DF=2). Error bars denote standard error of the mean measured across the mean values for each animal. Faded lines show percentage values for individual mice that contained hit, try, and run trials for a given trial type. (**F**) Odds ratio between receipt of reward for trials on which epochs contain task-anchored firing relative to trials on which epochs contain task-independent firing (beaconed-G, p=0.23; non-beaconed-G, p=0.001; probe-G, p=0.27).

The online version of this article includes the following figure supplement(s) for figure 6:

**Figure supplement 1.** Speed profiles across different trial outcomes.

**Figure supplement 2.** High correlation to the task-anchoring template of grid firing promotes localisation by path integration but is not required for cued localisation.

## Discussion

Our results support specific roles for task-anchored grid representations in path integration-dependent behaviours while arguing against the idea that grid codes provide a representation of location that is used more generally. Thus, we find that grid representations can either be anchored to position in a task environment, or can provide an environment-independent distance metric (*Figure 2*), that grid cells can switch between these operating modes within a behavioural session (*Figures 3–4*), and that anchoring of grid firing fields to location is not required for cued reward localisation, but appears to promote path integration-dependent reward localisation (*Figures 5–6*). By contrast, while some non-grid cells had activity that switched modes coherently with grid cells, many non-grid cells did not show

task-independent periodic activity (*Figure 3—figure supplement 5*, *Figures 4–5*, *Figure 5—figure supplement 2*), suggesting that the MEC may implement multiple, parallel spatial computations.

## Task-anchoring of grid cells varies within and between behavioural sessions

A standard view of grid firing is that it provides an 'always on' high capacity representation of current location. In contrast, our results demonstrate that grid representations are not necessarily anchored to the behaviourally relevant environmental reference frame. This extends previous observations that on circular tracks grid cells read out path integrated distance rather than absolute position (*Jacob et al., 2019*). Our results show that both modes of grid representation can be observed during the same behavioural task, and that the grid network can switch between these operating modes within a session. Consistent with the reported continuous attractor dynamics of grid networks, we find that simultaneously recorded grid cells switch modes coherently (*Figures 4 and 5*), although without recording from all grid cells simultaneously we cannot rule out the possibility that switching reflects a subset of grid cells that become disconnected from the wider network. The mode switching we observe here appears to differ from previously reported spontaneous remapping of MEC network states (*Low et al., 2021*) as the previously reported phenomenon was independent of any task contingencies and involved remapping between different task-anchored representations. By contrast, we show here switching between task-anchored representations that may be directly useful for solving the task at hand, and task-independent representations that appear unlikely to contribute to solving the task.

Grid cells are a relatively small proportion of the neurons in the MEC and an important question is the extent to which switches between task-anchored and task-independent modes extend to other cell types. Among non-grid cells, a sub-population appeared to switch between task-anchored and task-independent modes coherently with grid cells. However, task-independent periodic activity was relatively rare among non-grid cells (*Figure 3—figure supplement 5*). A possible explanation for this dissociation is that the MEC might contain multiple sub-networks, with populations of grid cells and coherent non-grid cells sharing continuous attractor dynamics, while other populations of non-grid cells operate independently from the grid network. For example, non-grid cells may be differentially influenced by visual cues (*Casali et al., 2018*; *Kinkhabwala et al., 2020*) or could generate location representations through ramping profiles rather than discrete firing fields (*Tennant et al., 2022*). The presence in the MEC of functionally distinct grid and non-grid networks might also account for why manipulations that perturb grid cells selectively disrupt path integration-dependent behaviours (*Gil et al., 2018*), whereas manipulations that target layer 2 stellate cells, which include grid and non-grid cells, impair path integration-dependent and cued behaviours (*Qin et al., 2018*; *Tennant et al., 2018*).

The causes of switching between task-anchored and task-independent representations may be an important focus for future investigation. It appears unlikely that task-independent coding by grid cells result from failure of upstream circuits to generate appropriate visual representations, e.g. through a shift in visual attention, as mice performed well on the visually cued version of the location memory task when grid cells were task-independent. A perhaps more promising hypothesis for future investigation is that switching reflects modulatory pathways reducing the impact of visual inputs to the grid system, possibly reflecting top-down control mechanisms, shifts in brain state, or uncertainty about whether visual or motor signals indicate the correct environment.

## Spatial roles of grid cells may be specific to path integration-dependent behaviours

Our finding that cued identification of a reward location is similar when grid cells are task-anchored or task-independent suggests that grid representations are not required for cued recall of locations. Thus, when both visual and grid inputs are available to downstream decision-making circuits, the grid input appears not to be used. If it was, then the inconsistent positional signals from the task-independent grid codes would impair performance. An implication of this result is that cue-rich tracks often used to investigate grid activity patterns may not engage behaviours that require anchored grid firing. Several observations suggest that the MEC could nevertheless be required for recall of cued locations. First, inactivation of stellate cells in layer 2 of the MEC causes deficits in the task we use here and in other visually cued tasks (*Qin et al., 2018*; *Tennant et al., 2018*). Second, other spatial

tasks that involve selections between cued locations appear to require the MEC (e.g. *Gaskin and White, 2013*; *Gaskin and White, 2010*). In this case, spatial representations used for the task could be encoded by other functional cell types, e.g. neurons that encode location through border (*Solstad et al., 2008*) or ramping firing fields (*Tennant et al., 2022*), or through cue-responsive cells (*Casali et al., 2018*; *Keene et al., 2016*; *Kinkhabwala et al., 2020*).

On trials in which the reward zone cue is absent, efficient reward localisation relies on path integration from the start of the track (*Tennant et al., 2018*). We found that on these trials, task-anchored grid firing was associated with a spatially localised stopping strategy and a higher proportion of successful trials compared with when grid firing was task-independent. This corroborates key predictions of hypothesised roles for grid cells in path integration (*McNaughton et al., 2006*). It is conceivable that behaviourally relevant computations are implemented elsewhere in the brain with grid anchoring to the track an indirect consequence, but explanations of this kind are hard to reconcile with evidence that stellate cells in the MEC are required for the task we used here (*Tennant et al., 2018*), or with evidence for specific roles of grid firing in path integration based on genetic manipulations that abolish grid firing without affecting other functional cell types (*Gil et al., 2018*). Nevertheless, our finding of residual localisation performance on task-independent trials suggests that additional neural mechanisms may also support path integration-dependent behaviour. This could reflect additional mechanisms for the implementation of path integration, e.g. through ramping activity (cf. *Tennant et al., 2022*). Alternatively, mice could in principle estimate track location with a system that utilises information about distance travelled obtained from task-independent grid representations. If multiple mechanisms support path integration then it will be important to establish when each contributes. For example, because grid representations are available on the first entry to an environment they may be important for behaviour in newly experienced locations, whereas for familiar locations complementary representational strategies that emerge with learning may be sufficient.

## Ideas and speculation

Our results point to a specific role for grid firing in path integration-dependent behaviour, and demonstrate the importance of anchoring of grid representations to task environments. One implication of our results is that rather than being an 'always on' tracking system, grid cell networks vary in their engagement with the environment. This may reflect control of the grid network by attentional or other top-down mechanisms. Alternatively, as the CA1 region of the hippocampus provides a major input to the MEC, instability of grid anchoring could be an indirect consequence of mechanisms that control the structure and stability of place cell maps (*Krishnan et al., 2022*; *Pettit et al., 2022*). In either case, our results motivate a focus on grid cell activity in tasks that require path integration, while an implication of our finding for investigations of grid cell activity using cue-rich environments is that in these experimental settings grid cells may not be influencing behavioural outcomes. Our results also offer a new perspective on interindividual differences in path integration by humans (e.g. *Chrastil et al., 2017*; *Lakshminarasimhan et al., 2018*; *Petzschner and Glasauer, 2011*). Thus, rather than resulting from variation in path integration per se, differences between individuals could instead result from variation in the anchoring of grid representations underlying path integration. This could be important as a potential mechanism for deficits in spatial localisation associated with neurological and neurodevelopmental disorders (*Kunz et al., 2015*; *Newton et al., 2023*; *Noel et al., 2020*).

## Materials and methods

### Key resources table

| Reagent type (species) or resource | Designation | Source or reference | Identifiers | Additional information |
| --- | --- | --- | --- | --- |
| Strain, strain background (mouse) | C57BL/6 | In-house breeding | NA | NA |
| Software, algorithm | R | NA | 4.2.3 | https://www.r-project.org/ |
| Software, algorithm | Python | NA | 3.8.1 | https://www.python.org/ |
| Software, algorithm | ImageJ | Fiji | NA | https://fiji.sc |

*Continued on next page*

*Continued*

| Reagent type (species) or resource | Designation | Source or reference | Identifiers | Additional information |
| --- | --- | --- | --- | --- |
| Software, algorithm | Blender | NA | 2.7 | https://www.blender.org/ |
| Software, algorithm | Open Ephys | NA | 0.4.4 | https://open-ephys.org/ |
| Software, algorithm | CTAn | NA | 1.13.5.1 | NA |
| Other | EIB-16 | NeuralLynx | Cat# 31-0603-0106 | NA |
| Other | Platinum/Iridium wire | NeuralLynx | NA | NA |
| Other | Gold Plating Solution | | 20 ml Gold Plating Solution | NA |
| Other | Headpost | RIVETS | NA | https://dudmanlab.org/html/rivets.html |
| Other | UV curing dental cement | RelyX | Cat# 56874 | https://www.3m.co.uk/3M/en_GB/p/d/b00007450/ |
| Other | Simplex Rapid | Kemdent | Cat# ACR803 | https://www.kemdent.co.uk/simplex-rapid-powder-clear-225g?osCsid=j0b5160aallnl2kcdtjpas4oj1 |
| Other | Omnetics to Mill-Max adaptor | Axona | HSADPT-NN1 | NA |
| Other | RHD 6 ft ultrathin SPI cable | Intan | Cat# C3206 | https://open-ephys.org/ |
| Other | RHD 6 ft standard SPI cable | Intan | Cat# C3218 | https://intantech.com/ |
| Other | Acquisition board | Open Ephys | NA | https://open-ephys.org/ |

All experiments were carried out under a UK Home Office project licence, were approved by the Animal Welfare and Ethical Review Board of the University of Edinburgh College of Medicine and Veterinary Medicine, and conformed with the UK Animals (Scientific Procedures) Act 1986 and the European Directive 86/609/EEC. Nine male C57BL/6J mice were used in this study. Three of the nine mice used here were also part of a previous study (*Tennant et al., 2022*).

## Microdrive fabrication and surgical procedures

Microdrive fabrication and surgical procedures were similar to our previous work (*Gerlei et al., 2020*; *Tennant et al., 2022*). Microdrives containing four tetrodes were built by threading 90% platinum, 10% iridium tetrode wires (18 µm HML-coated, Neuralynx) to an EIB-16 board (Neuralynx) via an inner cannula (21 gauge 9 mm long). The board was covered in epoxy and a poor lady frame (Axona) cemented to the side. An outer cannula (17 gauge 7 mm), placed around the inner cannula, was secured temporarily using vaseline, allowing it to be lowered during the surgery. Tetrodes were trimmed to ~3 mm using ceramic scissors (Science Tools, Germany) and gold-plated (Non-cyanide Gold Plating Solution, Neuralynx) to give an impedance between 150 and 200 kΩ at 1 kHz.

Before surgery, tips of the tetrodes were washed with ethanol and then sterile saline. Anaesthesia was induced using 5% isoflurane/95% oxygen, and sustained at 1–2% isoflurane/98–99% oxygen. After exposing the surface of the skull a RIVETS headpost (*Osborne and Dudman, 2014*) was attached to the skull with UV curing resin cement (RelyX Unicem, 3M). For electrical grounding, two M1 ×4 mm screws (AccuGroup SFE-M1-4-A2) were implanted through small craniotomies drilled on the left hemisphere ~3.4 mm lateral, and ~1 mm rostral relative to Bregma and the centre of the intraparietal plate, respectively. The microdrive was attached to a stereotaxic frame via an Omnetics to Mill-Max adaptor (Axona, HSADPT-NN1) and the tetrodes lowered 1.2–1.4 mm into the right hemisphere of the brain, beginning at 3.4 mm lateral from Bregma and along the lambdoid suture, and at an angle of –15 degrees in the posterior direction. The outer cannula was lowered and sealed with sterile vaseline, and the implant fixed to the skull with UV curing resin. After the resin hardened, the grounding wires were wrapped around the grounding screws and fixed with silver paint (RS components 101-5621). The grounding screws were covered with resin and any holes filled with dental cement (Simplex Rapid). After the surgery, mice recovered

for ~20 min on a heat mat, had unlimited access to Vetergesic jelly (0.5 mg/kg of body weight buprenorphine in raspberry jelly) for 12 hr, and before proceeding were given a minimum of 2 days postoperative recovery.

## Behavioural and electrophysiological recording

The behavioural setup, training procedures, and recording approaches were similar to those described previously (*Tennant et al., 2022*; *Tennant et al., 2018*). Mice were handled twice a day for 7 days following surgery. They were then habituated to the virtual reality setup for 10–20 min per day over 2 consecutive days. After each habituation session the mice were given access to soy milk to familiarise them with the reward and were given access to an open arena for 5–10 min of free exploration. From 4 to 5 days before starting training their access to food was restricted so that their body weight was ~85% of its baseline value, calculated from its weight prior to restriction and normalised to the expected daily growth for the animal's age.

Experimental days involved recording from mice in the open arena and then in the virtual location memory task. On a few days this order was reversed without apparent effects on the results obtained. Mice were collected from the holding room 30–60 min before recording, were handled for 5–10 min, weighed and placed for 10–20 min in a cage containing objects and a running wheel. Between recording sessions mice were placed back in the object-filled playground for 30 min. Tetrodes were typically lowered by 50–100 µm after each session. The open arena consisted of a metal box with a square floor area, removable metal walls, metal frame (Frame parts from Kanya UK, C01-1, C20-10, A33-12, B49-75, B48-75, A39-31, ALU3), and an A4-sized cue card in the middle of one of the metal walls. For the open-field exploration session, mice were placed in the open arena while tethered via an ultrathin SPI cable and custom-build commutator and left unprompted for 30 min to freely move around. For the location memory task mice were trained to obtain rewards at a location on the virtual linear track. Mice were head-fixed using a RIVETS clamp (Ronal Tool Company, Inc) and ran on a cylindrical treadmill fitted with a rotary encoder (Pewatron). Virtual tracks, generated using Blender3D ( blender.com) had length 200 cm, with a 60 cm track zone, a 20 cm reward zone, a second 60 cm track zone, and a 60 cm black box to separate successive trials. The distance visible ahead of the mouse was 50 cm. The reward zone was either marked by distinct vertical green and black bars on 'beaconed' trials, or was not marked by a visual cue at all on 'non-beaconed' or 'probe' trials. A feeding tube placed in front of the animal dispensed soy milk rewards (5–10 µl per reward) if the mouse stopped in the reward zone, however was not dispensed on probe trials. A stop was registered in Blender3D if the speed of the mouse dropped below 4.7 cm/s. Speed was calculated on a rolling basis from the previous 100 ms at a rate of 60 Hz.

Trials were delivered in repeating blocks throughout a recording session. For example, three beaconed trials (B) followed by two non-beaconed trials (N) with the order repeating until the end of the session. To encourage learning and engagement in both beaconed and non-beaconed trials, the first day of training typically used a trial type ratio of three beaconed trials to one non-beaconed trials. As training progressed we then increased the proportion of non-beaconed trials up to a ratio of one beaconed trial to four non-beaconed trials. Examples of trial blocks include BBBBN, BBBN, BBN, BN, BNN, BNNN, BBNNN, and BBNNNNNNN where each character indicates the trial type of each trial within a block. In some sessions we replaced single non-beaconed trials in trial blocks with a probe trial (P). Examples of trial blocks with probe trials include BBBBNBBBBP, BBBNBBBP, BBNBBP, and BBNNNNNNNP. The ratio and order of trial type delivery was not found to affect the results obtained (*Figure 3—figure supplement 7*).

Electrophysiological signals were acquired using an Intan headstage connected via an SPI cable (Intan Technologies, RHD2000 6 ft [1.8 m] standard SPI interface cable) attached to an Open Ephys acquisition board. Signals were filtered (2.5–7603.8 Hz). For the location memory task, behavioural variables including position, trial number, and trial type were calculated in Blender3D at 60 Hz and sent via a data acquisition (DAQ) microcontroller (Arduino Due) to the OpenEphys acquisition board. In the open arena, motion and head-direction tracking used a camera (Logitech B525, 1280×720 pixels Webcam, RS components 795-0876) attached to the frame. Red and green polystyrene balls were attached to the sides of the headstage and were tracked using a custom script written in Bonsai (*Lopes and Monteiro, 2021*). Synchronisation of position and electrophysiology data used an LED attached to the side of the open arena in the field of view of the camera, with randomly timed trigger

pulses sent to the LED via an Arduino board (Arduino Uno) and to the Open Ephys acquisition board via the I/O board.

Following experiments, tetrodes were localised using a microCT scanner (*Source data 1 and 2*). Mice were anaesthetised with isoflurane and then a lethal dose of sodium pentobarbital (Euthatal, Meridal Animal Health, UK) were perfused with a mixture of PFA and glutaraldehyde, and the head with the microdrive still intact on the skull left in the same solution for two nights. All tissue and bone except that attached to the microdrive was removed before washing the brains in ddH$_2$O and incubating at 4°C for 2 weeks in 2% osmium tetroxide (2% OsO$_4$). Brains were then washed in ddH$_2$O, dehydrated in ethanol and then embedded in resin. After the resin had cured the brains were imaged in a microCT scanner (Skyscan 1172, Bruker, Kontich, Belgium). Scanning parameters were: source voltage 54 kV, current 185 μA, exposure 885 ms with a 0.5 mm aluminium filter between the X-ray source and the sample. The scan dataset was reconstructed (CTAn software, v1.13.5.1) and viewed with DataViewer (Bruker). Tetrodes were localised relative to landmarks in version 2 of the Allen Reference Atlas for the mouse brain (https://mouse.brain-map.org/static/atlas) (see *Source data 1 and 2*).

## Spike sorting

Spikes were isolated from electrophysiological data using an automated pipeline based on MountainSort (v0.11.5 and dependencies) (*Chung et al., 2017*; *Gerlei et al., 2020*). Recordings from the open-field and virtual reality tasks were concatenated for spike sorting. Pre-processing steps converted Open Ephys files to mda format, filtered signals between 600 and 6000 Hz, and performed spatial whitening over all channels. Events were detected from peaks >3 standard deviations (SD) above baseline and separated by at least 0.33 ms from other events on the same channel. The first 10 principal components of the detected waveforms were used as inputs to the ISOSPLIT algorithm. Cluster quality was evaluated using isolation, noise overlap, and peak signal-to-noise ratio metrics (*Chung et al., 2017*). Units with firing rate >0.2 Hz, isolation >0.9, noise overlap <0.05, and peak signal-to-noise ratio >1 were used for further analysis. Downstream analyses were carried out using Python (v3.8.1) and R (v4.2.3).

## Analysis of neural activity in the open arena

For analysis of neural activity in the open arena, firing rate maps were calculated by binning spikes into 2.5 cm bins, dividing by the total time occupied in each bin and then smoothed with a Gaussian kernel. Autocorrelograms were calculated by sliding the rate map over all x and y bins and calculating a correlation score. Grid scores were defined as the difference between the minimum correlation coefficient for rate map autocorrelogram rotations of 60 and 120 degrees and the maximum correlation coefficient for autocorrelogram rotations of 30, 90, and 150 degrees (see *Sargolini et al., 2006*). Fields were detected in the autocorrelogram by converting it into a binary array using 20% of the maximal correlations as a threshold. If the binary array had more than seven local maxima, a grid score was calculated. Correlations between the rotated autocorrelograms were then calculated using a ring containing the six local maxima closest to the centre of the binary array and excluding the maximum at the centre. The ring was detected based on the average distance of the six fields near the centre of the autocorrelogram (middle border=1.25 * average distance, outer border=0.25 * average distance). To compute the spatial stability of cells in the open arena, the within-session spatial correlation was calculated by computing the Pearson correlation between the firing rate map from the first half session and the second half session. Bins that were not visited in both halves were excluded from the calculation. Neurons were classified as grid cells when their grid score and spatial stability score was in the 99th percentile of the same scores from 1000 shuffled datasets. Shuffled spike data was generated by drawing a single value from a uniform distribution between 20 and 580 s and adding this to the timestamp of each spike. Spike times that exceeded the recording duration were wrapped to the start of the session. Spike locations were recomputed from the shuffled spike times and spatial scores recalculated.

## Analysis of behaviour during the location memory task

Plots of running speed as a function of location on the virtual track were generated by first binning speed into 1 cm location bins for each trial and then smoothing by convolution with a Gaussian filter (SD=2 cm, SciPy Python package).

Trials were classified into hits and misses based on whether the mouse stopped (speed <4.7 cm/s) within the reward zone. Miss trials were further split by comparing their average speeds in the reward zone to hit trials; the 95th percentile of speeds in the reward zone for hit trials was used to discriminate between try trials (<95th percentile speed) and run trials (>95th percentile speed; *Figure 5—figure supplement 1*). Trials in which the mouse's average speed outside of the reward zone was <10 cm/s were left unclassified.

To compute stop density profiles (e.g. *Figure 5C–F*), stops were counted within 1 cm location bins and the counts were divided by the number of trials to obtain the number of stops per cm per trial. This was smoothed by convolution with a Gaussian filter (SD=1 cm). To evaluate the stop density when aggregating sessions or for epochs of trials within a session (e.g. *Figure 6A and C*), the same procedure was applied to first generate a stop density profile for the trials of interest within a session. We then subtracted an average of density profiles calculated in the same way for shuffled data from the same trials but in which stop locations were randomly drawn from a uniform distribution of track locations. In this way, stop densities below zero in the subtracted profiles can be interpreted as below chance relative to their average shuffled distributions, and stop densities greater than zero can be interpreted as greater than chance. Aggregate stop density profiles were then generated by averaging the individual subtracted profiles. Where average stop density plots were shown for coding epochs (e.g. *Figure 6C*, *Figure 6—figure supplement 2E*), these were weighted both on the proportion of trials classified in a particular coding scheme (e.g. a code weight of 0.05 when 5% of trials were classified as task-anchored for a single grid cell) and the number of simultaneously recorded grid cells such that single sessions were weighted equally (e.g. a session weight of 0.2 [1/5] when five grid cells are simultaneously recorded in a session). Similar weighted averages were also applied to plots where the proportion of hit, try, and run trials differed within a session in a similar vein (*Figure 6E*, *Figure 6—figure supplement 2G*).

## Analysis of neural activity during the location memory task

Firing rate maps for each trial were generated by dividing the track into 1 cm bins, summing the number of spikes in each bin, and dividing by the time the animal spent there. Firing rates were smoothed with a Gaussian filter (SD =2 cm). Spatial information was calculated in bits per second as

$$\sum_{i=1}^{N} P_i \lambda_i \log_2 \left( \frac{\lambda_i}{\lambda} \right)$$

where i indexes a position bin in the firing rate map, N is the number of bins, $P_i$ is the occupancy probability, $\lambda_i$ is the firing rate in the bin, and $\lambda$ is the mean firing rate (*Skaggs and McNaughton, 1996*). When spatial information scores were generated for epochs within a session, we took the same number of trials for each epoch (e.g. when comparing task-anchored and task-independent epochs). This was done by randomly subsetting the epoch with the greater number of trials to match the number of trials of the epoch with the smaller number of trials.

To quantify the spatial periodicity of neural firing, the Lomb-Scargle method of least squares spectral analysis (LSSA) as implemented by the Astropy Python module (*Price-Whelan et al., 2022*) was used to generate a frequency spectrum in the spatial domain (*Lomb, 1976*; *Scargle, 1982*). A periodogram was computed every 10 cm with a sample length equal to three track lengths (600 cm). Track locations were normalised between 0 and 1 so that spatial frequencies corresponded to the number of oscillations per trial. Spatial frequencies >5 were discarded from further analysis as no grid cells were found with grid spacings <40 cm. Individual periodograms were combined to create an average periodogram across the session (*Figure 1—figure supplement 1* for an illustrative example).

To distinguish spatially periodic firing from aperiodic firing (*Figure 2*), we compared the spectral peaks from a cell's averaged periodogram to a false alarm threshold. The threshold was calculated with a bootstrapping method that used 1000 shuffled instances of the cell's firing, with the aim of disrupting spatial periodicity while preserving any local temporal firing. This was inspired by the field shuffle procedure used for grid cells in two-dimensional environments (*Barry and Burgess, 2017*). First, firing fields were identified by detecting the peaks and troughs in a smoothed version of the cell's firing rate map (convolution with a Gaussian filter, SD=4 cm) with a minimum peak distance of 20 cm, smaller peaks were removed until all conditions of the detection were satisfied using the SciPy function scipy.signal.find_peaks. Fields were defined as the region between adjacent troughs.

The field positions were reallocated in an unsmoothed rate map to random positions, preserving the spatial organisation of the field, while bins not attributed to a firing field filled the remaining gaps. The shuffled unsmoothed rate map was then smoothed by convolution with a Gaussian filter (SD=2 cm) and the periodogram calculated. This was repeated 1000 times and the 99th percentile calculated from the distribution of shuffled peak powers used to create the false alarm threshold. Cells with a measured peak power below this threshold were classified as aperiodic (*Figure 1—figure supplement 2* for an illustrative example).

To establish whether periodic cells had activity anchored to the track we calculated the difference between the peak spatial frequency of their periodogram and the nearest positive-integer value. We identified task-anchored periodic cells as those for which the difference was ≤0.05, and task-independent period cells as those for which the difference was >0.05. Using this metric yielded high prediction accuracies when comparing our classification to true labels when we simulated task-anchored or task-independent grid cells (*Figure 1—figure supplement 5*).

To classify activity within a recording session, we calculated average periodograms across a rolling window with a size of 200 individual periodograms, which equated to 10 trials (considering 20 samples per 200 cm track with 10 cm steps between periodogram increments). The peak of the average periodogram from a single window was identified, and the peak power and the spatial frequency at which this occurred extracted. To determine if the peak reflected a periodic signal, it was compared with an adjusted false alarm threshold (see below). Windows with periodograms containing peaks above the threshold were then classified as task-anchored or task-independent as for session-level periodograms, while other windows were classified as aperiodic. To classify at the level of individual trials, midpoints of each rolling window were extracted and assigned to trial numbers by assessing which trial number the midpoint was closest to. For example, a rolling window with a midpoint of 10.4 would be assigned to trial number 10, while a midpoint of 10.6 (or 10.5 in border cases) would be assigned to trial number 11. All rolling windows with their respective classifications from a single trial would be pooled and counted. The classification with the most counts for that trial number would then be assigned to that trial (*Figure 3—figure supplement 1* for an illustrative example). When assigning a common classification across a group of cells recorded simultaneously (e.g. *Figure 4B*), we used the mode of their classification, in cases where the group only contained two cells and there was not 100% agreement between these two cells, one cell was selected at random to represent the common population code.

To select the optimal rolling window size we considered how the false alarm threshold changes as a function of the number of periodograms used to compute the average periodogram (*Figure 3—figure supplements 1 and 2*). As the number of samples within the rolling window increases, the false alarm threshold decreases (*Figure 3—figure supplement 3*). To account for this, an adjusted false alarm threshold was calculated using the first 200 periodograms from a field shuffle to compute an average periodogram. This was repeated for 1000 field shuffles and the 99th percentile of the peak powers used as the adjusted false alarm threshold. We also found that the number of periodograms used to calculate the average periodogram greatly affected the prediction accuracy and bias. We opted for a rolling window size equal to 200 periodograms as this was found to achieve high accuracy with minimal bias (*Figure 3—figure supplement 2*).

To calculate the coding agreement between any two simultaneously recorded cells, the coding schemes were compared across the course of the session. The agreement score for a cell pair was equal to the percentage of trial classifications that agreed. In order to compare the measured agreement against chance for any cell pair, shuffled arrangements of trial classifications were generated for one of the two cells by splicing the trial classification raster where consecutive trials switch coding schemes and then reordering them at random (*Figure 4D*). This procedure was repeated 10 times for each cell pair and agreement scores calculated accordingly. Where shuffled agreement scores are visualised (*Figure 4E*), the average of the shuffled agreement scores are shown.

A potential weakness of the Lomb-Scargle method that we used for trial-level classification is that the window over which classification is made has limited resolution, while simulations on artificial data suggested that classification could be biased towards task-independent firing depending on how frequent transitions between coding schemes occur. We therefore implemented a second method for trial-level classification (*Figure 6—figure supplement 2*). With this method we first created an average firing rate profile of the task-anchored trials obtained from the Lomb-Scargle method. We

then used this average as a template which we correlated with the firing rate profile of each trial. For analyses in *Figure 6—figure supplement 2*, trials with a correlation coefficient ≥0.5 were then classified as task-anchored positive (TA+) and task-anchored negative (TA-) otherwise. We note that when using this method we discarded units for which less than 15% of trials were originally classified as task-anchored, as for these units we were unable to generate templates of sufficient quality.

## Simulation of firing during the location memory task

To evaluate classification of grid firing as task-anchored or task-independent, we first simulated various cell types including grid cells, place cells, ramp cells, and aperiodic cells (*Figure 1—figure supplement 4*). For each cell type, we simulated an agent moving with a constant velocity of 10 cm/s across 100 trials of a 200 cm long linear track and logged the locations visited with a sampling rate of 1000 Hz. For each cell type, the probability of firing at any given location was defined by a probability density function (PDF) with a range of 0–1. The average firing rate was set by multiplying this normalised PDF by a scalar variable $P_{max}$(spike) which by default was set to 0.1. Firing events were then assigned to each sampled location based on the scaled PDF. Firing rate maps and subsequent periodograms were created as described above.

PDFs for grid cells were created by positioning Gaussians kernels at equidistant locations along the track, with kernel SD equal to 0.1 multiplied by a given grid spacing between the kernels. To simulate task-anchored grid codes, the Gaussian kernels were positioned at the same track location on each trial, whereas to simulate task-independent grid codes the kernels were positioned independently from the track with distances equal to the grid spacing. To simulate field jitter, a displacement of the kernel position was drawn from another Gaussian distribution with mean=0 cm and SD=0 cm (for no jitter) or SD=10 cm (for default jitter). A random jitter was drawn for each field and was used to shift fields accordingly. The PDF for the place cell example was made up of a singular Gaussian kernel (mean=100 cm, SD=10 cm) positioned at the centre of the track and was repeated every trial. The PDF for the ramp cell example consisted of a linear ramp from the start of the track (0 cm) to the end of the track (200 cm). The PDF for the 'random field' cell was created by first generating the PDF for the place cell example and then passing this through the field shuffle as described (see *Figure 1—figure supplement 2*). The PDF for the Gaussian noise example was a uniform distribution.

To generate PDFs of grid cells alternating between task-anchored and task-independent codes, representations of each type were generated and merged (*Figure 3—figure supplement 2*). Considering the rolling classification computes a prediction label across a number of trials, we reasoned the manner and frequency of the alternation would affect prediction. We simulated this alternation in both blocks of trials and at the level of single trials. For the simulations that alternated in blocks of trials, the initial trial was randomly assigned to either the task-anchored PDF or the task-independent PDF with equal probability. For all subsequent trials there was a 10% chance of alternating to the other PDF (e.g. task-anchored to task-independent or task-independent to task-anchored). For simulations that alternated at the level of single trials, every trial was randomly assigned to either the task-anchored PDF or the task-independent PDF, with equal probability.

To evaluate our classification of periodic firing at the level of individual cells, we simulated 500 task-anchored grid cells and 500 task-independent grid cells with grid spacings uniformly distributed between 40 and 400 cm and compared the true labels of these simulated cells with the predicted classifications (*Figure 1—figure supplement 5*). To determine what spatial frequency tolerance to use in our classification, we classified the simulated dataset across the full range of spatial frequency thresholds. This was repeated using a range of $P_{max}$(spike) and jitter SD values. To simplify the analysis, no field shuffle was computed on simulated data and therefore no false alarm threshold was used. This effectively forced our classifier to label cells as task-anchored or task-independent without the possibility of an aperiodic label. The prediction accuracy was calculated as the percentage of true task-anchored and task-independent coding grid cells with a correct prediction label. The prediction bias was calculated as the percentage of actual task-anchored cells minus the percentage of predicted task-anchored cells. As the classification in the simulation analysis had only two valid labels, positive bias represents over classification as task-independent whereas negative bias represents over classification as task-anchored. In plots showing prediction bias, positive and negative bias is relabeled to reflect this.

To evaluate classification of periodic firing at the level of individual trials, we simulated 100 grid cells with grid spacings uniformly distributed between 40 and 400 cm that could alternate between task-anchored and task-anchored task-independent coding task-independent either in blocks of trials or every trial (see above) and compared the true labels of these simulated cells and trials with the predicted classifications (*Figure 3—figure supplement 2*). To determine how many periodograms to average over (or if any), we classified the simulated dataset across a range of rolling window sizes to map at what rolling window size we could maximise prediction accuracy and minimise bias between task-anchored and task-independent classifications. Again, this was repeated using a range of $P_{max}$-(spike) and jitter SD values and no field shuffles were computed. The prediction accuracy of our classification was calculated as the average percentage of true task-anchored and task-independent coding trials with a correct prediction label. The prediction bias was calculated as the average percentage of actual task-anchored trials minus the percentage of predicted task-anchored trials across cells.

## Statistical analyses

Group comparisons used linear mixed effect models (*Figures 2F, 3G, 4E–F, Figure 3—figure supplement 4*) or generalised linear mixed effect models (*Figures 4E, 6B and D–F*) implemented using lme4 (*Bates et al., 2015*), lmerTest (*Kuznetsova et al., 2017*), and glmmTMB (*Brooks et al., 2017*) packages within R (*R Development Core Team, 2021*), with model comparisons using ANOVA and post-fitting pairwise comparisons (*Searle et al., 1980*) using estimated marginal means (*Lenth et al., 2024*). Fits of firing rate, spatial information, and peak width in *Figures 2F and 3G* and *Figure 3—figure supplement 4A-B, D* used log transformed data. Fits in *Figure 4E* (comparisons of shuffled data) used a beta family function, and in *Figure 6* used a binomial family function with logit linker. Random effects had a nested structure to account for animals and sessions (all models), and where appropriate neuron identity (*Figures 4E and 6*). For analyses in *Figure 6*, trials classified as 'aperiodic' were removed from the dataset to facilitate direct comparison of trials with task-anchored and task-independent aperiodic grid firing. To estimate the effect size of the relative influence of task-anchored firing versus task-independent firing on task performance (*Figure 6F*), odds ratios and confidence intervals were extracted from the full model using sjPlot (*Lüdecke et al., 2023*).

## Acknowledgements

We thank Caswell Barry for helpful discussions. This work was supported by the Wellcome Trust (200855/Z/16/Z to MFN), MRC Precision Medicine PhD programme (MR/S502479/1 to HC), and the Simons Initiative for the Developing Brain (to MFN). This work made use of resources provided by the Edinburgh Compute and Data Facility. For the purpose of open access, the author has applied a CC BY public copyright license to any author accepted manuscript version arising from this submission.

## Additional information

### Funding

| Funder | Grant reference number | Author |
|---|---|---|
| Medical Research Council | MR/S502479/1 | Harry Clark |
| Wellcome Trust | https://doi.org/10.35802/200855 | Matthew F Nolan |
| Simons Initiative for the Developing Brain | | Matthew F Nolan |

For the purpose of Open Access, the authors have applied a CC BY public copyright license to any Author Accepted Manuscript version arising from this submission.

### Author contributions

Harry Clark, Conceptualization, Data curation, Formal analysis, Investigation, Visualization, Methodology, Writing - original draft, Writing - review and editing; Matthew F Nolan, Conceptualization,

Formal analysis, Supervision, Writing - original draft, Project administration, Writing - review and editing

### Author ORCIDs
Harry Clark http://orcid.org/0000-0001-9843-0276
Matthew F Nolan http://orcid.org/0000-0003-1062-6501

### Ethics
All experiments were carried out under a UK Home Office project licence, were approved by the Animal Welfare and Ethical Review Board of the University of Edinburgh College of Medicine and Veterinary Medicine, and conformed with the UK Animals (Scientific Procedures) Act 1986 and the European Directive 86/609/EEC.

Reviewer #1 (Public Review): https://doi.org/10.7554/eLife.89356.3.sa1
Reviewer #2 (Public Review): https://doi.org/10.7554/eLife.89356.3.sa2
Author Response https://doi.org/10.7554/eLife.89356.3.sa3

## Additional files

### Supplementary files
- Supplementary file 1. Summary table of recorded cells and estimated tetrode locations.
- MDAR checklist
- Source data 1. MicroCT imaging for tetrode localisation 1/2.
- Source data 2. MicroCT imaging for tetrode localisation 2/2.

### Data availability
Data have been deposited at Edinburgh DataShare and code at GitHub (copy archived at Zenodo).

The following datasets were generated:

| Author(s) | Year | Dataset title | Dataset URL | Database and Identifier |
| --- | --- | --- | --- | --- |
| Clark and Nolan | 2024 | Task-anchored grid cell firing is selectively associated with successful path integration-dependent behaviour | https://datashare.ed.ac.uk/handle/10283/8723 | Edinburgh DataShare, 10.7488/ds/7684 |
| Clark HD | 2024 | MattNolanLab/eLife_Grid_anchoring_2024: v1.0.0.0 | https://doi.org/10.5281/zenodo.10698051 | Zenodo, 10.5281/zenodo.10698051 |

The following previously published dataset was used:

| Author(s) | Year | Dataset title | Dataset URL | Database and Identifier |
| --- | --- | --- | --- | --- |
| Clark et al. | 2022 | Spatial representation by ramping activity of neurons in the retrohippocampal cortex | https://datashare.ed.ac.uk/handle/10283/4498 | Edinburgh DataShare, 10.7488/ds/3515 |

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
