## [Editor Report · eLife assessment]

This **valuable** study examines the relationship between positional anchoring of grid cell activity and performance in spatial navigation tasks that require path integration. The authors demonstrate that grid cells can either fire in relation to the position relative to task-relevant virtual stimuli or independently based on the distance covered. Their findings **convincingly** reveal that mice exhibited better performance in the path integration task when grid cell activity was anchored to their position on the virtual track rather than the distance traversed, highlighting the contribution of grid firing to spatial navigation behavior. The work will be of interest to experimental and computational neuroscientists interested in spatial navigation.

---

## [Referee Report · Reviewer #1 (Public Review)]

Summary:

In this study, Clark et. al. used electrophysiology approaches to measure MEC neuron activity while mice performed spatial memory tasks in one-dimensional virtual tracks, where the mice must stop in a specific reward zone for a reward. The authors identified that grid cell activity could either be anchored to the track reference frame ('task-anchored') or can maintain a periodic firing pattern independent of the track reference frame ('task-independent'). They found that in the task that requires path integration, good task performance is specifically associated with task-anchored grid cell activity.

Strength:

This study took advantage of the variation in neural activity and navigation task behaviors to answer an important question: how grid cell activity is associated with performance of spatial tasks. The mice performed individual trials where they must stop in a specific reward zone for a reward. Individual behavioral sessions could include three types of trials: (1) a visual cue at the reward location (beaconed trials), (2) no cue at the reward location (non-beaconed trials), and (3) no cue and no reward regardless of stopping (probe trials). The authors found that, interestingly, grid cell activity pattern could be anchored to task reference frame or maintain a periodic pattern independent of the reference frame. The anchoring of activity patterns could switch within a behavioral session. On the other hand, spatial firing of non-grid cells was either coherent with the grid population or was stably anchored to the task reference frame. Combining grid cell activity feature with task behaviors, they uncovered an association between the task-anchoring of grid cell activity with good performance in spatial navigation tasks that requires path integration (non-beaconed and probe trials). This work suggests the contribution of grid firing to path integration-dependent navigation.

Weakness:

It would be interesting to find out that on the trial-by-trial basis, whether the activity anchoring switched first, or the task behaviors altered first, or whether they happened within the same trial. This will potentially determine whether the encoding is causal for the behavior, or the other way around. However, based the authors explanation, their experimental design lacks sufficient statistical power to address the timing of mode switches within a trial, because task mode switching is relatively infrequent (so the n for switching is low) and only a subset of trials are uncued (making the relevant n even lower), while at a trial level the behavioral outcome is variable (increasing the required n for adequate power).

In addition, the authors reported that the activity anchoring of some non-grid cells coherently switched with grid cells, while others do not. They propose that the MEC implement multiple coding schemes. However, it is unclear whether and how the coding scheme is associated with behavior. It would be interesting to further investigate this question.

---

## [Referee Report · Reviewer #2 (Public Review)]

Clark and Nolan's study aims to test whether the stability of grid cell firing fields is associated with better spatial behavior performance on a virtual task. Mice were trained to stop at a rewarded location along a virtual linear track. The rewarded location could be marked by distinct visual stimuli or be unmarked. When the rewarded location was unmarked, the animal had to estimate its distance run from the beginning of the trial to know where to stop. When the mouse reached the end of the virtual track, it was teleported back to the start of the virtual track.

The authors found that grid cells could fire in at least two modes. In the "task-anchored" mode, grid firing fields had stable positions relative to the virtual track. In the "task-independent" mode, grid fields were decoupled from the virtual cues and appeared to be located as a function of distance run on the track. Importantly, on trials in which the rewarded location was unmarked, the behavioral performance of mice was better when grid cells fired in the "task-anchored" mode. When a unique visual cue marked the reward location, navigation performance was not correlated with the grid cells' firing mode.

This study is very timely as there is a pressing need to identify/delimit the contribution of grid cells to spatial behaviors. More studies are needed in which grid cell activity is linked to navigational abilities. The link proposed by Clark and Nolan between "task-anchored" coding by grid cells and navigational performance is a significant step toward better understanding how grid cell activity might support behavioral behavior. The results also highlight that some forms of navigation (approaching a location marked by a visual cue) might be less dependent on the anchoring of grid cells.

It should be noted that the study by Clark and Nolan is correlative. Therefore, the effect of selective manipulations of grid cell activity on the virtual task will be needed to evaluate whether the activity of grid cells is causally linked to the behavioral performance on this task. A previous study by the same research group showed that inactivating the synaptic output of stellate cells of the medial entorhinal cortex affected mice's performance of the same virtual task (Tennant et al., 2018). Although this manipulation likely affects non-grid cells, it is still one of the most selective manipulations of grid cells that are currently available.

It is interesting to consider how grid cells remain anchored to virtual cues. Recent work shows that grid cell activity spans the surface of a torus (Gardner et al., 2022). A run on the track can be mapped to a trajectory on the torus. Assuming that grid cell activity is updated primarily from self-motion cues on the track and that the grid cell period is unlikely to be an integer of the virtual track length, having stable firing fields on the virtual track likely requires a resetting mechanism taking place on each trial. During this resetting event, the active location on the torus is likely to jump to a new toroidal location, independently of self-motion cues. Future studies in which large numbers of grid cells are recorded could pinpoint at which moment such resetting event occurs on each trial.

---

## [Author Response · Author 
response]

The following is the authors’ response to the original reviews.

We thank the editors and reviewers for their tremendously helpful comments. We outline below changes we have made to the manuscript in response to each point. These include new analyses and a substantial rewrite to address the concerns about lack of clarity.

We believe the revisions strengthen the evidence for our conclusion that grid fields can be either anchored to or independent from a task reference frame, and that anchoring is selectively associated with successful path integration-dependent behaviour. Our additional analyses of non-grid cells indicate that while some are coherent with the grid population, many are not, suggesting cell populations within the MEC may implement grid-dependent and grid-independent computations in parallel.

We hope the reviewers will agree that our novel experimental strategy complements and avoids limitations of perturbation-based approaches, and by providing evidence to dissociate the two major hypotheses for whether and when grid cells contribute to behaviour our results are likely to have a substantial impact on the field.

**Public Reviews:**

**Reviewer #1 (Public Review):**
In this study, Clark et. al. uncovered an association between the positional encoding of grid cell activity with good performance in spatial navigation tasks that requires path integration, highlighting the contribution of grid firing to behaviour… The conclusions of this paper are mostly well supported by data, the finding about the association between grid cell encoding and behaviour in spatial memory tasks is important. However, some aspects of the analysis need to be clarified or extended.

Thank you for the overview and constructive comments.

(1) While the current dataset aims to demonstrate a "correlation" between grid cell encoding and task performance, the other variables that could confound this correlation should be carefully examined.(1.1) The exact breakdown of the fraction of beaconed/non-beaconed/probe trials is never shown. if the session makeup has a significant effect on the coding scheme or other results, this variable should be accounted for.

The lack of information about the trial organisation was a substantial oversight in our preparation of the first version of the manuscript. Session make up can not account for effects on grid stability and its relationship to behavioural outcome but this was not made at all clear.

In all sessions trial types were varied in a fixed repeating sequence. Therefore, continuous blocks of trials on which grid firing is anchored (or independent from) the track can not be explained by the mouse experiencing a particular trial type. We have revised the manuscript to make this clearer, e.g. p 5, ‘These switches could not be explained by variation between trials in the availability of cues or rewards, as these were interleaved in blocks that repeated throughout a session (see Methods), whereas periods in which grid cell activity was in a given mode extended across the repeating blocks (e.g. Figures 3D,E, 4A, 5E,F).’ and methods p 12, ‘Trials were delivered in repeating blocks throughout a recording session…’

(1.2) The manuscript did not provide information about whether individual mice experienced sessions with different combinations of the three trial types, and whether they show different preferences in position or distance encoding even in comparable sessions. This leads to the question of whether different behaviour and activity encoding were dominated by experimental or natural differences between individual mice. Presenting the data per mouse will be helpful.

As we note above, because trial types were interleaved in a fixed sequence, experience of a particular trial type can not account for switching between task-anchored and taskindependent firing modes. This was insufficiently clear in the first version of the manuscript.

We varied the proportions of trials of a particular type between sessions with the aim of maximising the number of non-beaconed and probe trials. This was necessary because we find that if we introduce too high a proportion of these trials early in training then mice appear to ‘lose interest’ in the task and their performance drops off. We therefore used an approach in which we increased the proportions of non-beaconed and probe trials over training days as mice became familiar with the task. This is now described in the methods (p 12).

Because the decision for when to vary the proportion of trial types was based on the previous day’s performance, the experimental design was not optimised for addressing the reviewer’s question about dissociating experimental from natural differences in mice. To provide some initial insight we have analysed the relationship between task anchored coding and proportion of beaconed trials in a session (Figure 3, Figure Supplement 7). While on average there is a higher proportion of trials in which grid fields are task-anchored in sessions with more beaconed trials, this effect is small and most of the variance is independent from the proportion of beaconed trials.

(1.3) Related to the above point, in Figure 5, the mice appeared to behave worse in probe trials than non-beaconed trials. If the mouse did not know if a trial is a probe or a non-beacon trial, they should behave equivalently until the reward location and thus should stop an equal amount. If this difference is because multiple probe trials are placed consecutively, did the mouse learn that it will not get a reward and then stop trying to get rewards? Did this affect switching between position and distance coding?

Thankyou for flagging this. This reflected an inconsistency arising from the way we detected stops that we have now corrected. Briefly, the temporal resolution of the processed location data against which the stop detection threshold was applied was insufficiently high. As a result, stops in the non-beaconed group were picked up, as they tended to be longer because mice remained still to consume rewards, whereas some stops in the probe group were missed because they were relatively short. We have corrected this by repeating the analyses on raw position data at the highest temporal resolution available. This analysis is now clearly described in the Methods (see p13 “A stop was registered in Blender3D if the speed of the mouse dropped below 4.7 cm/s. Speed was calculated on a rolling basis from the previous 100 ms at a rate of 60 Hz.”).

(1.4) It is not shown how the behaviours (e.g., running speed away from the reward zone, licking for reward) in beaconed/non-beaconed/probe trials were different and whether the difference in behaviours led to the different encoding schemes.

Because trial types were interleaved and repeated with a period less than the length of typical trial sequences during which grid cell activity remained either task-anchored or taskindependent, differences between trial types are unlikely to explain use of the different coding schemes. Hopefully, this is clarified by the comments above.

To further describe the relationship between behavioural outcomes, trial types and grid anchoring, we now also show running speed as a function of location for each combination of trial types and trial outcomes (Figure 6, Figure Supplement 1). This illustrates and replicates our previous findings (Tennant et al. 2018) that running speed profiles are similar for a given trial outcome regardless of trial type (Figure 6, Figure Supplement 1A), and further further shows that the behavioural profile for a given trial outcome and trial-type does not differ when grid cells are in task-anchored and task-independent modes (Figure 6, Figure Supplement 1B). This further argues against the possibility that difference in behaviours leads to the different encoding schemes.

(2) Regarding the behaviour and activity encoding on a trial-by-trial basis, did the behavioural change occur first, or did the encoding switch occur first, or did they happen within the same trial? This analysis will potentially determine whether the encoding is causal for the behaviour, or the other way around.

This is a good question but our experimental design lacks sufficient statistical power to address the timing of mode switches within a trial. This is because mode switching is relatively infrequent (so the n for switching is low) and only a subset of trials are uncued (making the relevant n even lower), while at a trial level the behavioural outcome is variable (increasing the required n for adequate power).

(3) The author determined that the grid cell coding schemes were limited to distance encoding and position encoding. However, there could be other schemes, such as switching between different position encodings (with clear spatial fields but at different locations), as indicated by Low et. al., 2021, and switching between different distant encodings (with different distance periods). If these other schemes indeed existed in the data, they might contribute to the variation of the behaviours.

Switching between position encoding schemes appears to be rare within our dataset and unlikely to contribute to variation in behaviour. In most sessions we did not observe switching between grid phases / position encodings (e.g. Figures 2A-B, 3B-E, 4A, 5C-D, F). In one session we found switching between different phases when grid cells were taskanchored. Because the grid period was unchanged, the spatial periodograms remained similar. We report this example in the revised manuscript (Figure 5E).

(4) The percentage of neurons categorised in each coding scheme was similar between nongrid and grid cells. This implies that non-grid cells might switch coding schemes in sync with grid cells, which would mean the whole MEC network was switching between distance and position coding. This raises the question of whether the grid cell coding scheme was important per se, or just the MEC network coding scheme.

We very much appreciate this suggestion. We note first that while the proportion of taskanchored grid and non-grid cells is similar, task-independent periodic firing of non-grid cells is much rarer than for grid cells (Figure 2E), suggesting a dissociation between the populations. To further address the question we have included additional analyses of nongrid cells (Figure 3, Figure Supplement 5). This shows that while some non-grid cells have anchoring that switches coherently with simultaneously recorded grid cells, others do not. Figures 4 and 5 now show examples of non-grid cell activity recorded simultaneously with grid cells.

Together, our data suggest that the MEC implements multiple coding schemes: one that is associated with the grid network and includes some non-grid cells; and one (or more) that can be independent from the grid network. This dissociation adds to the insights into MEC function that are provided by our study and is now highlighted in the abstract and discussion.

(5) In Figure 2 there are several cell examples that are categorised as distance or position coding but have a high fraction of the other coding scheme on a per-trial basis. Given this variation, the full session data in F should be interpreted carefully, since this included all cells and not just "stable" coding cells. It will be cleaner to show the activity comparison only between the stable cells.

We have now included examples in Figure 2A-C where the grid mode is stable throughout a session. As the view of activity at a session level is important, we have not updated Figure 2F, but have clarified the terminology to now clearly refer to classification at either season or trial levels. In addition, we have repeated the analyses shown in Figure 2F but after grouping cells according to whether their firing has a single mode on >85% of the trials (Figure 3 Figure Supplement 4). This analysis supports similar conclusions to those of Figure 2F.

(6) The manuscript is not well written. Throughout the manuscript, there are many unexplained concepts (especially in the introduction) and methods, mis-referenced figures, and unclear labels.

We very much appreciate the feedback and have substantially rewritten the manuscript. We have paid particular attention to explaining key concepts in the introduction and have carefully checked the figures. We welcome further feedback on whether this is now clearer.

**Reviewer #2 (Public Review):**
Clark and Nolan's study aims to test whether the stability of grid cell firing fields is associated with better spatial behaviour performance on a virtual task… This study is very timely as there is a pressing need to identify/delimitate the contribution of grid cells to spatial behaviours. More studies in which grid cell activity can be associated with navigational abilities are needed.

Thank you for the supportive comments and highlighting the importance of the question.

The link proposed by Clark and Nolan between "virtual position" coding by grid cells and navigational performance is a significant step toward better understanding how grid cell activity might support behaviour. It should be noted that the study by Clark and Nolan is correlative. Therefore, the effect of selective manipulations of grid cell activity on the virtual task will be needed to evaluate whether the activity of grid cells is causally linked to the behavioural performance on this task. In a previous study by the same research group, it was shown that inactivating the synaptic output of stellate cells of the medial entorhinal cortex affected mice's performance of the same virtual task (Tennant et al., 2018). Although this manipulation likely affects non-grid cells, it is still one of the most selective manipulations of grid cells that are currently available.

Again, thank you for the supportive comments. We recognise the previous version of the manuscript did not sufficiently clarify the motivation for our approach, or the benefits of capitalising on behavioural variable variability as a complementary strategy to perturbation approaches. We now make this clearer in the revised introduction (p 2, paragraphs 2 and 3).

When interpreting the "position" and "distance" firing mode of grid cells, it is important to appreciate that the "position" code likely involves estimating distance. The visual cues on the virtual track appear to provide mainly optic flow to the animal. Thus, the animal has to estimate its position on the virtual track by estimating the distance run from the beginning of the track (or any other point in the virtual world).

We appreciate the ambiguity here was confusing. We have re-named the groups to ‘taskanchored’, corresponding to when grid cells encode position on the track (as well as distance as the reviewer correctly points out), and ‘task-independent’, corresponding to the group we previously referred to as distance encoding.

It is also interesting to consider how grid cells could remain anchored to virtual cues. Recent work shows that grid cell activity spans the surface of a torus (Gardner et al., 2022). A run on the track can be mapped to a trajectory on the torus. Assuming that grid cell activity is updated primarily from self-motion cues on the track and that the grid cell period is unlikely to be an integer of the virtual track length, having stable firing fields on the virtual track likely requires a resetting mechanism taking place on each trial. The resetting means that a specific virtual track position is mapped to a constant position on the torus. Thus, the "virtual position" mode of grid cells may involve (1) a trial-by-trial resetting process anchoring the grid pattern to the virtual cues and (2) a path integration mechanism. Just like the "virtual position" mode of grid cell activity, successful behavioural performance on non-beaconed trials requires the animal to anchor its spatial behaviour to VR cues.
**Reviewer #3 (Public Review):**
This study addresses the major question of 'whether and when grid cells contribute to behaviour'. There is no doubt that this is a very important question. My major concern is that I'm not convinced that this study gives a significant contribution to this question, although this study is well-performed and potentially interesting. This is mainly due to the fact that the relation between grid cell properties and behaviour is exclusively correlative and entirely based on single cell activity, although the introduction mentions quite often the grid cell network properties and dynamics. In general, this study gives the impression that grid cells exclusively support the cognitive processes involved in this task. This problem is in part related to the text.

Thank you for the comments. We recognise now that the previous text was insufficiently clear. We have modified the introduction to clarify the value of an approach that takes advantage of behavioural variability. Importantly, this approach is complementary to perturbation strategies we and others have used previously. In particular it addresses critical limitations of perturbation strategies which can be confounded by off-target effects and possible adaptation, both of which are extremely difficult to fully rule out. We hope that with this additional clarification it is now clear that as for any important question multiple and complementary testing strategies are required to make progres, and second, that our study makes a new and important contribution by introducing a novel experimental approach and by following this up with careful analyses that clearly distinguish competing hypotheses.

However, it would be interesting to look at the population level (even beyond grid cells) to test whether at the network level, the link between behavioural performance and neural activity is more straightforward compared to the single-cell level. This approach could reconcile the present results with those obtained in their previous study following MEC inactivation.

We’re unclear here about what the reviewer means by ‘more straightforward’ as clear relationships between activity of single grid cells and populations of grid cells are well established (Gardner et al., 2021; Waaga et al., 2021; Yoon et al., 2013).

To give a clearer indication of the corresponding population level representations, as mentioned in response to Reviewer #1, we now include additional data showing many simultaneously recorded neurons, and analyses of non-grid as well as grid cells (Figures 4, 5, Figure 5 Figure Supplement 2).

To reconcile results with our previous study of MEC inactivation we have paid additional attention to the roles of non-grid cells (following suggestions by Reviewer #1). We show that while some non-grid cells show transitions between task-anchored and task-independent firing that are coherent with the grid population, many others have more stable firing that is independent of grid representations. This is consistent with the idea that the MEC supports localised behaviour in the cued and uncued versions of the task (Tennant et al., 2018), and suggests that while grid cells preferentially contribute when cues are absent, non-grid cells could also support the cued version. We make this additional implication clear in the revised abstract and discussion.

The authors used a statistical method based on the computation of the frequency spectrum of the spatial periodicity of the neural firing to classify grid cells as 'position-coding' (with fields anchored to the virtual track) and 'distance-coding' (with fields repeating at regular intervals across trials). This is an interesting approach that has nonetheless the default to be based exclusively on autocorrelograms. It would be interesting to compare with a different method based on the similarities between raw maps.

While our main analyses use a periodogram-based method to identify when grid cells are / are not anchored to the task environment, we validate these analyses by examination of the rate maps in each condition (Figures 2-4). For example, when grid cells are task-anchored, according to the periodogram analysis, the rate maps clearly show spatially aligned peaks, whereas when grid cells are not anchored the peaks in their rate maps are not aligned (Figure 2A vs 2B; Figure 3B-E; Figure 4C). We provide further validation by showing that spatial information (in the track reference frame) is substantially higher when grid cell activity is task-anchored vs task-independent (Figures 2F, 3G, 4F and Figure 3 Figure Supplement 4).

To further address this point we have carried out additional complementary analyses in which we identify task anchored vs task independent modes using a template matching method applied to the raw rate maps (Figure 6, Figure Supplement 2). These analyses support similar conclusions to our periodogram-based analyses.

Beyond this minor point, cell categorization is performed using all trial types.Each trial type (i.e. beacon or non-beacon) is supposed to force mice to use different strategies and should induce different spatial representations within the entorhinal-hippocampal circuit (and not only in the grid cell system). In that context, since all trials are mixed, it is difficult to extrapolate general information.

We recognise that the description of the task design was insufficiently clear but are unsure why ‘it is difficult to extrapolate general information’. Before addressing this point, we should first be clear that mice are not ‘forced’ to adopt any particular strategy. Rather, on uncued trials a path integration strategy is the most efficient way to solve the task. However, mice could instead use a less efficient strategy, for example by stopping at short intervals they still obtain rewards. Detailed behavioural analyses indicate that such random stopping strategies are used by naive mice, while with training mice learn to use spatial stopping strategies (Tennant et al. 2018).

In terms of ‘extracting general information’ from the task, the following findings lead to general predictions: (1) Grid cells can exist in either task-anchored or task-independent periodic firing modes; (2) These modes can be stable across a session, but often modeswitching occurs within a session; (3) While some non-grid cells show task-independent periodic firing, this is much less common than for grid cells, which suggests a model in which many non-grid MEC neurons operate independently from the grid network; (4) When a marker cue is available mice locate a reward equally well when grid cells are in taskanchored versus task-independent modes, which argues against theories in which grid cells are a key part of a general system for localisation; (5) When markers cues are absent taskanchored grid firing is associated with successful reward localisation, which corroborates a key prediction of theories in which grid cells contribute to path integration.

In revising the manuscript we have attempted to improve the writing to make these advances clearer, and have clarified methodological details that made interpretation more challenging than it should have been. For example, as noted in our response to Reviewer #1, we have included additional details to clarify the organisation of trials and relationships between trials, behavioural outcomes and neural codes observed.

On page 5 the authors state that 'Since only position representations should reliably predict the reward location, ..., we reasoned that the presence of positional coding could be used to assess whether grid firing contributes to the ongoing behaviour'. I do not agree with this statement. First of all, position coding should be more informative only in a cue-guided trial. Second, distance coding could be as informative as position coding since at the network level may provide information relevant to the task (such as distance from the reward).

Again, this point perhaps reflects a lack of clarity on our part in writing the manuscript. When grid cells are anchored to the track reference frame (now called ‘tasked anchored’, previously ‘position encoding’), then the location of the rate peaks in grid firing is reliable from trial to trial. This is the case whether or not the trial is cued. When grid cells are independent of the track reference frame (now called ‘task independent’, previously ‘distance encoding’), then the location of the firing rate peaks vary from trial to trial. In the latter case, position can not be read out directly from trial to trial.

In principle, in the task-independent mode track position could be calculated by storing the grid network configuration at the start of the track, which would differ on each trial, and then implementing a mechanism to readout relative distance as mice move along the track. However, if mice do use this computation we would expect them to do so equally well on cued and uncued trials. By contrast, our results clearly show a dissociation between trial types in the relationship between grid firing and behavioural outcome. We highlight and discuss this possibility in the revised manuscript (p 10, ‘Alternatively, mice could in principle estimate track location with a system that utilises information about distance travelled obtained from task-independent grid representations’).

Third, position-coding is interpreted as more relevant because it predominates in correct trials. However, this does not imply that this coding scheme is indeed used to perform correct trials.

We have revised the manuscript to clarify our goal of distinguishing major hypotheses for the roles of grid cells in behaviour (Introduction, ‘On the one hand, theoretical arguments that grid cell populations can generate high capacity codes imply that they could in principle contribute to all spatial behaviours Fiete et al., 2008; Mathis et al., 2012; Sreenivasan and Fiete, 2011). On the other hand, if the behavioural importance of grid cells follows from their hypothesised ability to generate position representations by integrating self-motion signals (McNaughton et al., 2006), then their behavioural roles may be restricted to tasks that involve path integration strategies.’

By showing that performance on cued trials is similar regardless of whether grid cells are task-anchored or not, we provide strong evidence against the idea that grid firing is in general necessary for location-based behaviours. By showing that task anchoring is associated with successful localisation when cues are absent we corroborate a key prediction of hypothesised roles for grid cells in path integration-dependent behaviour. Therefore, we substantially reduce the space of behaviours to which grid cells might contribute. Importantly, this space is much larger for the MEC, which is required for cued and uncued versions of the task. We have revised the introduction and discussion to make these points clearer.

While we believe our results add a key piece of evidence to the puzzle of when and where grid cells contribute to behaviour, we agree that further work will be required to develop and test more refined hypotheses. Alternative models also remain plausible, for example perhaps the behaviourally relevant computations are implemented elsewhere in the brain with grid anchoring to the track as an indirect consequence. Nevertheless, explanations of this kind are more difficult to reconcile with evidence that inactivation of stellate cells in the MEC impairs learning of the task, and other manipulations that modify grid firing impair performance on similar tasks. We now discuss these possibilities (discussion p 10, ‘mice could in principle estimate track location with a system that utilises information about distance travelled obtained from task-independent grid representations’).

It could be more informative to push forward the correlative analysis by looking at whether behavioural performance can be predicted by the coding scheme on a trial-by-trial basis.

The previous version of the manuscript showed these analyses (now in Figure 6). Thus, task anchored grid firing predicts more successful performance on uncued trials at the session level (Figure 6A-B) and at the trial level (Figure 6C-D).

**Reviewer #1 (Recommendations For The Authors):**
(1) The author particularly mentioned that the 1D tracks are different from the "cue-rich environments that are typically used to study grid cells". It is not clear what conclusions would hold for a cue-rich environment or a track, which may require relatively less path integration compared to the cue-sparse environment. This point should be discussed.

This is an important point that we did not pay sufficient attention to in the previous version of the manuscript. Our finding of successful localisation in the cued environment when grid cells are not task anchored implies that grid anchoring is not required to solve cued tasks. The implication here is that cue rich environments may then not be the most suitable for investigation of grid roles in behaviour as non-grid mechanisms may suffice, although this does not rule out the possibility that anchored grid codes may play important roles in learning about cue rich environments. We now address this point in the discussion (p 10, ‘An implication of this result is that cue rich tracks often used to investigate grid activity patterns may not engage behaviours that require anchored grid firing.’).

(2) It would be good to see the statistics for the number of different cells (stable position or distance encoding, and unstable cells) identified per mouse/session and the number of grid cells per session.

These are now added to Supplemental Data 2 and will also be accessible through code and datasets that we will make available alongside the version of record.

(3) Figure 2F: any explanation about why AG cells had high spatial information?

Previously the calculation used bits per spike and as aperiodic cells have low firing rates the spatial information was high. We have replaced this with bits per second, which provides a more intuitive measure and no longer implies high spatial information. We have amended this in the methods (p 15, ‘Spatial information was calculated in bits per second…’).

(4) The following methods sections should provide additional details:(4.1) Details of the training protocol are largely left to reference papers. The reference papers give a general outline of the training protocol, but the details are not completely comparable given the single experiment performed on these mice. More details should be given on training stages and experience at the time of the experiment.

The task is more clearly described in the introduction (p 3), and additional details of the training protocol are now provided in the methods (p 12-13).

(4.2) The methods reference mean speed across sessions, but it is not clear where this was used.

This was very poor wording. We have now changed this to ‘For each session the mean speed was calculated for each trial outcome’.

(4.3) The calculation of the spatial autocorrelogram on a per-trial basis should be more explicitly stated. Is it the average of each 10 cm increment with the centre trial?

We have added additional information to the methods (p 16-18).

(4.4) 1D field detection is not sufficiently explained in Figure 1/S2. This information should also appear in the methods section.

This is now clarified on page 16 in section ‘Analysis of neural activity and behaviour during the location memory task’.

(5) The data in Figure 4A and B only shows speed vs. location for one example mouse. The combined per mouse or per session data should also be shown.

This is now shown in Figure 5A and Figure 5, Figure Supplemental 2

(6) Figure 5 is somewhat confusing. Why are A/B by session and C/D by trial? The methods imply that A/B are originally averaged by cell, but that duplicate cells in the same session are excluded because behaviour versus session type is identical. This method should be valid if all grid cells within a session are all "stable". This is likely given the synchrony of code-switching between grid cells, but not all co-active grid cells behaved identically.It is understandable that C/D are performed by trial, but it should be made clear that it is not a comparable analysis to A/B. It is unclear what N refers to in C. The figure says by trial, but the legend says the error bar is by cell. If data is calculated by trial and then averaged by cell, this should be more clearly stated.

In Figure 6A/B (previously Figure 5A/B) we focus our analysis on sessions in which the mode of grid firing, either task-anchored or task-independent, was relatively stable on a trialto-trial basis (see Figure 3F for definitions). This enables us to then compare behaviour averaged across each session, with sessions categorised as task-anchored and task independent. This analysis has the advantage that it focuses on large blocks of time (whole sessions) in which the mode of grid firing is unambiguous, but the disadvantage is that it excludes many sessions in which grid firing switches between task-anchored and taskindependent modes.

Figure 6C/D (previously Figure 5C/D) addresses this limitation by carrying out similar analyses with behaviour sorted into task-anchored versus task-independent groups at the level of trials. A potential limitation for this analysis is that grid firing is somewhat variable on a trial-by-trial basis and so some trials may be mis-classified. We don’t expect this to lead to systematic bias, but it may make the data more noisy. Nevertheless, these analyses are important to include as they allow assessment of whether conclusions from 6A/B hold when all sessions are considered.

We have added additional clarification of the rationale for these analyses to the main text (p7-8, ‘’We addressed this by using additional trial-level comparisons’). We have also added clarification in the methods section for categorisation of task-anchored versus taskindependent trials when multiple grid cells were recorded simultaneously (p 17, ‘When assigning a common classification across a group of cells recorded simultaneously...’) and an explanation for the N in the figure legend. We also clarify that the analyses use a nested random effects design to account for dependencies at the levels of sessions and mice (methods, p 20, ‘Random effects had a nested structure to account for animals and sessions…’) .

(7) Panels E and F of Figure 5 are not explained in the main text.

This is now corrected (see p8, ‘Additional analyses…’).

(8) Figure 5: Since stable grid cells and all grid cells are shown, it will be better to show unstable cells, which can be compared with grid cells.

Given that the rationale for differences between Figure 6A/B and C/D (previously Figure 5AD) were not previously clear, the reason for focussing on stable grid cells here was likely also not clear (see point 6 above). We don’t show unstable grid cells in Figure 6A-B as the behaviour averaged at the level of a session would be a mix of trials when they are taskanchored and when they are task-independent. Therefore, the analysis would not test predictions about the relationship between task-anchored vs task-independent modes and behaviour. We hope this is now clear in the manuscript given the revisions introduced to address point 6 above.

(9) The methods describing the statistics for these experiments are also confusing. The methods section should be written more clearly, and it should be made clear in the text or figure legend whether this data is the "original" data or is processed in relation to the model, such as excluding duplicate grid cells within a session. The figure legend should also state that a GLMM was used to calculate the statistics.

We have revised the methods section with the goal of improving clarity, adding detail and removing ambiguity. This includes updates of the methods for the GLMM analysis, which are referred to within the Figure 6 legend. A clear definition of a stable session is now also added to the Figure 6 legend.

**Reviewer #2 (Recommendations For The Authors):**
When grid fields are anchored to the virtual world (position mode), there is probably small trialto-trial variability in the firing location of the firing fields. Is this trial-to-trial variability related to the variability in the stop location? This would provide a more direct link between path integration in grid cell networks and behaviour that depends on path integration.

When attempting to address this we find that the firing of individual grid cells is too variable to allow sufficiently precise decoding of their fields at a single trial level. This is expected given the Poisson statistics of spike generation and previous evaluations of grid coding (e.g. (Stemmler et al., 2015)).

The conclusion of the abstract is: "Our results suggest that positional anchoring of grid firing enhances the performance of tasks that require path integration." This statement is slightly confusing. The task requires (1) anchoring the behaviour to the visual cues presented at the start of the trial and (2) path integration from thereon to identify the rewarded location. The performance is higher when grid cells anchor to the visual cues presented at the start of the trial. What the results show is that the anchoring of grid firing fields to visual landmarks enhances the performance of tasks that require path integration from visual landmarks (i.e. grid cells being anchored to the reference frame that is behaviorally relevant).

To try to more clearly explain the logic and conclusion we have rewritten the abstract, including the final sentence.

Similar comment for the title of Figure 5: "Positional grid coding is not required for cued spatial localisation but promotes path integration-dependent localisation." Positional coding means that grid cells are anchored to the behaviorally relevant reference frame.

To address the lack of clarity we have modified the little of Figure 6 (previously Figure 5) to read ‘Anchoring of grid firing to the task reference frame promotes localisation by path integration but is not required for cued localisation’.

In Figure 1, there is a wide range of beaconed (40-80%) and non-beaconed (10-60%) trials given. It is not 100% clear whether these refer to the percentage of trials of a given type within the recording sessions. Was the proportion of non-beaconed trials manipulated? If so, was the likelihood of position and distance coding changing according to the percentage of nonbeaconed trials?

The ranges given refer to proportions across different behavioural sessions. Within any given behavioural session the proportion was constant. We now make this clear in the figure legend and in the results and methods sections.

We did not manipulate proportions of trial types during a session. Manipulations betweens sessions were carried out with the goal of maximising the numbers of uncued trials that the mice would carry out (see response to public comments above). While the effect of trial-type at the session level is not relevant to the hypotheses we aim to test here, we have included an additional analysis of the relationship between task anchoring and the proportions of trial types in a session (Figure 3, Figure Supplement 7)(also discussed above). As disentangling the effects of learning and motivation will be complex and likely require new experimental designs we have not drawn strong conclusions or pursued the analysis further..

I was not convinced that the labels "position" and "distance" were appropriate for the two grid cell firing modes. My understanding is that the "position" code also requires the grid cell network to estimate distance. It seems that the main difference between the "position" and "distance" modes is that when in the "position" mode, the activity on the torus is reset to a constant toroidal location when the animal reaches a clearly identifiable location on the virtual track. In the "distance" mode, this resetting does not take place.

As previously mentioned, we agree these terms weren’t the best and have since relabelled these as “task-anchored” and “task-independent”.

There are a few sections in the manuscript that implicitly suggest that a causal link between grid cell activity and behaviour was demonstrated. For instance: "It has been challenging to directly test whether and when grid cells contribute to behaviour.": The assumption here is that the manuscript overcomes this challenge, but the study is correlative.

We have modified the wording to be clear that we are introducing new tests of predictions made by hypotheses about causal relationships between grid coding and behaviour (introduction, p 1-2). We also clarify that our results argue against the hypothesis that grid cells provide a general coded for behaviour, but corroborate predictions of hypotheses in which they are specifically important for path integration (discussion, p 10).

We have modified the title abstract and main text to try to treat claims about causality with care. We now more thoroughly introduce and contrast the approach we report here with previous experiments that use perturbations (introduction, p2). While it is tempting to make stronger claims for causality with these approaches, there are also logical limitations with perturbation-based approaches, for example the challenges of fully excluding off target effects and adaptation. We now explain how these strategies are complementary. Our view is that both strategies will be required to develop strong arguments for whether and when grid cells contribute to behaviour. From this perspective, it is encouraging that our conclusions are in agreement with what are probably the most specific perturbations of grid cells reported to date (Gil et al. 2017), while perturbations that more generally affect MEC function appear to impair cued and path integration-dependent behaviours (Tennant et al. 2018). We now discuss these points more clearly (introduction, p 2).

I am slightly confused by the references to the panels in Figure 4."In some sessions, localization of the reward occurred almost exclusively when grid cells were anchored to position and not when they encoded distance (Figure 4C). Figure 4C only shows position coding."In other sessions, animals localised the reward when grid firing was anchored to position or distance, but overall performance was improved on positional trials (Figure 4D-E)." The reference should probably point to Figure 4E-F or just to 4E."In a few sessions, we observed spatial stopping behaviour comparable to cued trials, even when grid firing almost exclusively encoded distance rather than position (Figure 4F)." From Figure 4F, it seems that the performance on non-beaconed trials is better during "position" coding.

We have now updated Figure 5 (Figure 4 in the original manuscript) and references to the Figure in the text. Now Figure 5 shows the activity of cells recorded in stable and unstable task-anchored and task-independent sessions (see Figure 5C-F).

Minor issues:Is this correct: (Figure 4A and Figure 4, Figure Supplement 1).

This has been corrected.

Figure 4B: There could be an additional label for position and distance.

Figure 4B from the original manuscript has now been removed.

Figure 4C-F. The panels on the right side should be explained in the Figure Legend.

Legends for Figure 5C-F (previously Figure 4C-F) have now been updated.

**Reviewer #3 (Recommendations For The Authors):**
Specific questions :(1) Position coding reflects a coding scheme in which fields are spaced by a fixed distance; previous studies have shown that a virtual track grid map is a slice of the 2D classic grid. In that case, the fields are still anchored to the track but would produce a completely different map. Did the authors check whether it is the case at least for some cells? If not, what could explain such a major difference?

Το avoid confusion we now use the term ‘task-anchored’ rather than ‘position coding’ (see comments above). We should further clarify that our conclusions rest on whether or not the grid fields are anchored to the track. Task anchored firing does not require that grid fields maintain their spacing from 2D environments, only that fields are at the same track position on each trial. Thus, whether the spacing of the fields corresponds to a slice through a 2D grid makes no difference to the hypotheses we test here.

We agree that the relationship between 1D and 2D field organisation could be an interesting future direction, for example anchoring could involve resetting the grid phase while maintaining a stable period, or it could be achieved through local distortions in the grid period. However, since these outcomes would not help distinguish the hypotheses we test here we have not included analyses to address them.

(2) Previous studies have highlighted the role of grid cells in goal coding. Here there is an explicit reward in a particular area. Are there any grid modifications around this area? This question is not addressed in this study.

Again, we note that the hypotheses we test here relate to the firing mode of grid cells - taskanchored or task-independent - and interpretation of our results is independent from the specific pattern of grid fields on the track. This question nevertheless leads to an interesting prediction that if grid fields cluster in the goal area then this clustering should be apparent in the task-anchored but not the task-independent firing mode.

We test this by considering the average distribution of firing fields across all grid cells in each firing mode (Reviewer Figure 1). We find that when grid firing is task-anchored there is a clear peak around the reward zone, which is consistent with previous work by Butler et al. and Boccara et al. Consistent with our other prediction, this peak is reduced when grid cells are in the task-independent mode.

**Author response image 1. sa3fig1:** Plot shows the grid field distribution during stable grid cell session (> 85 % task-anchored or task-independent) (**A**) or during task-anchored and task-independent trials (**B**). Shaded regions in A and B represent standard error of the mean measured across sessions and epochs respectively.

(3) The behavioural procedure during recording is not fully explained. Do trial types alternate within the same session by blocks? How many trials are within a block? Is there any relation between trial alternation and the switch in the coding scheme observed in a large subset of the grid cells?

We agree this wasn’t sufficiently clear in the previous version of the manuscript. Trial types were interleaved in a fixed order within each session. We have updated the results and methods sections to provide details (see responses above).

(4) From the examples in Figure 2 it seems that firing fields tend to shift toward the start position. Is it the case in all cells? Could this reflect some reorganisation at the network level with cells signalling the starting as time progresses?

This is inconsistent between cells. To make this variability clear we have included additional examples of spiking profiles from different grid cells (Figure 2 - 5). Because quantification of the phenomena would not, so far as we can tell, help distinguish our core hypotheses we have not included further analyses here.

(5) Are grid cells with different coding properties recorded in different parts of the MEC? Are there any differences between these cell categories in the 2D map?

The recordings we made are from the dorsal region of the MEC (stated at the start of the results section). We don’t have data to speak to other parts of the MEC.

Minor:There are very few grid cell examples that repeat in the different figures. I would suggest showing more examples both in the main text and supplementary material.

We have now provided multiple additional examples in Figures 2, 4 and 5. Grid cell examples repeat in the main figures twice, in both cases only when showing additional examples are shown from the same recording session (Figure 2A example #1 with Figure 5C, Figure 3E with Figure 4A). Further similar repeats are found in the supplemental figures (Figure 3D with Figure 5, Figure Supplement 2A, Figure 3C with Figure 5, Figure Supplement 2F).

Fig1 A-B shows the predictions in a 1D track based on distance or position coding. The A inset represents the modification of field distribution from a 2D arena to a 1D track, as performed in this study. The inset B is misleading since it represents the modifications expected from a circular track to a 1D track as in Jacob et al 2019, that is not what the authors studied. It would be better to present either the predictions based on the present study or the prediction based on previous studies. In that case, they should mention the possibility that the 1D map is a slice of the 2D map.

The goal of Figure 1A-B is to illustrate predictions (right) based on conclusions from previous studies (left). Figure 1A shows predicted 1D track firing given anchoring to the environment typically observed in grid cell studies in 2D arenas. Figure 1B shows predicted 1D track firing given the firing shifting firing patterns observed by Jacob et al. in a circular 2D track. To improve clarity, we have modified the legend to make clear that the schematics to the right are predictions given the previous evidence summarised to the left. As we outline above, the critical prediction relates to whether the representations anchor to the track. Whether the 1D representation is a perfect slice isn’t relevant to the hypotheses tested and so isn’t included in the schematic (see comments above).